# SALSA2.0: The sectional aerosol module of the aerosol-chemistry-climate model ECHAM6.3.0-HAM2.3-MOZ1.0

Harri Kokkola[1], Thomas Kühn[1,2], Anton Laakso[1,3], Tommi Bergman[4], Kari E. J. Lehtinen[1,2], Tero Mielonen[1], Antti Arola[1], Scarlet Stadtler[5], Hannele Korhonen[6], Sylvaine Ferrachat[7], Ulrike Lohmann[7], David Neubauer[7], Ina Tegen[8], Colombe Siegenthaler-Le Drian[9], Martin G. Schultz[5,10], Isabelle Bey[9,11], Philip Stier[12], Nikos Daskalakis[13], Colette L. Heald[14], and Sami Romakkaniemi[1]

[1]Atmospheric Research Centre of Eastern Finland, Finnish Meteorological Institute, P.O.Box 1627, FI-70211 Kuopio, Finland.

[2]Aerosol Physics Research Group, University of Eastern Finland, P.O.Box 1627, FI-70211 Kuopio, Finland.

[3]Department of Soil, Water and Climate, University of Minnesota, Twin Cities, St. Paul, MN 55108, USA

[4]Weather and Climate Models, Royal Netherlands Meteorological Institute, P.O.Box 201, 3730AE De Bilt, the Netherlands

[5]Institut für Energie- und Klimaforschung, IEK-8, Forschungszentrum Jülich, Germany

[6]Climate Research, Finnish Meteorological Institute, Helsinki, FI-00100, Finland

[7]Institute for Atmospheric and Climate Science, ETH Zurich, 8092 Zurich, Switzerland

[8]Modeling of Atmospheric Processes, Leibniz Institute for Tropospheric Research (TROPOS), Leipzig, Germany

[9]Centre for Climate Systems Modeling (C2SM), ETH Zürich, Switzerland

[10]Jülich Supercomputing Centre, JSC, Forschungszentrum Jülich, Germany

[11]Centre météorologique de Genève, Office fédéral de météorologie et de climatologie MétéoSuisse, av. de la Paix 7bis, CH-1211 Genève 2, Switzerland

[12]Department of Physics, University of Oxford, Parks Road, OX1 3PU, UK

[13]Laboratory for Modeling and Observation of the Earth System (LAMOS), Institute of Environmental Physics (IUP), University of Bremen, Bremen, Germany

[14]Department of Civil and Environmental Engineering, Department of Earth, Atmospheric, and Planetary Science, Massachusetts Institute of Technology, Cambridge, MA 02139

*Correspondence to:* Harri Kokkola (harri.kokkola@fmi.fi)

**Abstract.** In this paper, we present the implementation and evaluation of the aerosol microphysics module SALSA2.0 in the framework of the aerosol-chemistry-climate model ECHAM-HAMMOZ. It is an alternative microphysics module to the default modal microphysics scheme M7 in ECHAM-HAMMOZ. The SALSA2.0 implementation within ECHAM-HAMMOZ is evaluated against observations of aerosol optical properties, aerosol mass, and size distributions, comparing also to the skill of the M7 implementation. The largest differences between the implementation of SALSA2.0 and M7 are in the methods used for calculating microphysical processes, i.e. nucleation, condensation, coagulation, and hydration. These differences in the microphysics are reflected in the results so that the largest differences between SALSA2.0 and M7 are evident over regions where the aerosol size distribution is heavily modified by the microphysical processing of aerosol particles. Such regions are, for example, highly polluted regions and regions strongly affected by biomass burning. In addition, in a simulation of the 1991 Mt Pinatubo eruption in which a stratospheric sulfate plume was formed, the global burden and the effective radii of the stratospheric aerosol are very different in SALSA2.0 and M7. While SALSA2.0 was able to reproduce the observed time evolution of the global burden of sulfate and the effective radii of stratospheric aerosol, M7 strongly overestimates the

removal of coarse stratospheric particles and thus underestimates the effective radius of stratospheric aerosol. As the mode widths of M7 have been optimized for the troposphere and were not designed to represent stratospheric aerosol the ability of M7 to simulate the volcano plume was improved by modifying the mode widths decreasing the standard deviations of the accumulation and coarse modes from 1.59 and 2.0, respectively, to 1.2 similar to what was observed after the Mt Pinatubo eruption. Overall, SALSA2.0 shows promise in improving the aerosol description of ECHAM-HAMMOZ and can be further improved by implementing methods for aerosol processes that are more suitable for the sectional method, e.g size dependent emissions for aerosol species and size resolved wet deposition.

## 1 Introduction

Describing the global physical and chemical properties of the atmospheric aerosol in atmospheric models is challenging due to their large spatial and temporal variability. The diameter of the particles spans several orders of magnitude and the chemical composition can include hundreds of compounds (e.g., Colbeck et al., 2014). For example, when the nanometer sized smallest partices grow in size, they contribute to the number of aerosol particles which can form cloud droplets (Kulmala and Kerminen, 2008) while the largest particles of micrometer size can also affect rain formation (Jensen and Lee, 2008) Particles of different sizes affect both atmospheric radiation (Chung et al., 2005) and cloud processes (Lohmann and Feichter, 2005) in different ways (Boucher et al., 2013; Myhre et al., 2013). Therefore, in order to accurately simulate the effects of aerosol on the global climate, the entire aerosol particle size spectrum must be represented. In addition to the particle size, the chemical composition of particles, in particular the absorption (Dubovik et al., 2002) and solubility/hygroscopicity (Che et al., 2016) vary strongly between different aerosol constituents, influencing their ability to affect radiation and cloud interactions. In order to properly simulate these aerosol effects, the composition should also be adequately represented in the models.

This multitude of variability in the physical and chemical properties of aerosols poses a challenge for global modellers to describe aerosol particles in a computationally efficient way. Simulating the aerosol size distribution at high resolution including size resolved chemical composition within hundreds of thousands of grid boxes is computationally challenging. However, solving the size-resolved evolution of atmospheric particles computationally efficiently is not a new challenge as such simulations were made in the early years of computational atmospheric physics (e.g., Young, 1974). Currently, most of the global models which describe the evolution of the aerosol size distribution resort to using either modal or sectional approaches or a mix of these two (e.g., Mann et al., 2014). The application of sectional models in global 3-D simulations can involve a trade-off with horizontal or vertical resolution because sectional models are computationally more expensive.

Essentially, modal and sectional approaches can be considered as two variants of the same method, as both approaches divide the aerosol size distribution into size classes. The modal approach assumes individual size classes (modes) to be log-normally distributed and the total aerosol size distribution to be a superposition of these modes (e.g., Vignati et al., 2004; Stier et al., 2005). In the sectional approach, the size classes are either assumed to be monodisperse (Zaveri et al., 2008) , they are assumed to have a linear size distribution within a section (Young, 1974; Stevens et al., 1996) or a piecewise log-normal approximation within a section is used (von Salzen, 2006). The modal setup is usually computationally more efficient since the number of size

classes needed to represent typically observed size distributions is much smaller than in the sectional approach. Typically modal models use seven or fewer modes while sectional models use up to 100 size classes (Mann et al., 2014; Yu and Luo, 2009). On the other hand, sectional models allow for more flexibility in e.g. the shape of the size distribution and volume distribution of chemical compounds (Kokkola et al., 2009). Although sectional models have been shown to perform significantly better than

modal models in 0-D and 2-D frameworks (Weisenstein et al., 2007; Kokkola et al., 2009; Korhola et al., 2014) the benefits of sectional models in global 3-D simulations are less evident (Mann et al., 2012, 2014). It is also difficult to quantify the benefit of the sectional approach because the comparison between modal and sectional models are, in most cases, not done within the same model framework and the structural differences in the models cause such a large difference in the modeled aerosol that the contribution to the differences from the choice of the size distribution scheme can not be identified (Mann et al., 2014).

Another reason is that the evaluation of the skill of global aerosol models against observations is extremely challenging as the model value for a given observable may not represent the measured value at a particular monitoring site (Schutgens et al., 2016). This discrepancy can for example be caused by the fact that the global model value represents the mean for a grid box $\sim 200\,\mathrm{km} \times 200\,\mathrm{km}$ in size. Aerosol properties can exhibit large variations within that area and the measurement site may not represent the mean conditions within that grid box.

Here we present the implementation of the sectional aerosol microphysics module SALSA (Kokkola et al., 2008) in the aerosol-chemistry-climate model ECHAM-HAMMOZ (echam6.3-ham2.3-moz1.0) which also includes the modal aerosol microphysics module M7 (Vignati et al., 2004). The paper is structured as follows. In Section 2 we present the details of individual model components, especially the methods for solving aerosol processes. In Section 3 we briefly present the model to be analyzed with the different models/configurations. In Section 4 we present the evaluation of the model against observations. The

performance of the model is evaluated using retrievals of aerosol optical properties from both satellite and ground based remote sensing instruments. We also compare the model with in-situ observations, including vertical profiles of aerosol composition and mass from aircraft measurements. Finally, we compare the sectional model results with those obtained from ECHAM-HAMMOZ in modal aerosol configuration. The ECHAM-HAMMOZ model framework allows for running simulations in an otherwise very similar global model setup, but only switching between the modal and sectional aerosol representations. This

comparison provides insights on the impacts of the representation of the aerosol size distribution on the simulated aerosol properties, and thus on the simulated climate and climate effects.

## 2   Model description.

### 2.1   ECHAM

The host atmospheric model in ECHAM-HAMMOZ is the sixth generation atmospheric general circulation model ECHAM6.

The details of the model have been described by Stevens et al. (2013). It is the atmospheric component of the Max Planck Institute for Meteorology Earth System Model (MPI-ESM) and was originally based on the European Centre for Medium Range Weather Forecasts (ECMWF) weather prediction model (Simmons et al., 1989). The dynamical core applies the spectral method for calculating the atmospheric circulation and flux form semi-Lagrangian transport scheme. In our model configura-

tion, we use the T63 spectral truncation for the horizontal grid, with 47 flexible vertical levels which follow the terrain and use the hybrid vertical coordinate representation described in detail by Roeckner et al. (2003).

In atmosphere-only simulations, ECHAM6 uses prescribed sea surface temperatures (SST) and sea ice cover (SIC). The land processes are calculated using the JSBACH -model (Raddatz et al., 2007) which is integrated in ECHAM6. The aerosol processes are simulated by the HAMMOZ aerosol-chemistry model (Schultz et al., 2017).

## 2.2 HAMMOZ

The aerosol-chemistry model HAMMOZ combines the Hamburg Aerosol Model (HAM) and the Model for OZone And Related chemical Tracers (MOZART) which simulates the chemical reactions of 300 species through 650 reactions in the troposphere and stratosphere. A more detailed description of MOZ and its implementation in ECHAM-HAMMOZ is given in the accompanying paper by Schultz et al. (2017). Please note that in the simulations made for this paper, we did not use MOZ in any of the simulations. Instead, sulfate chemistry is calculated in the more simplified scheme of HAM (Zhang et al., 2012).

HAM will also be presented in detail in another accompanying paper by Tegen et al. (2018). However, as SALSA is integrated within HAM and as SALSA incorporates many of the model design characteristics of HAM, we briefly introduce the aerosol related features of HAMMOZ and detail the coupling between HAMMOZ and SALSA.

The HAM aerosol model has been designed to simulate all tropospherically relevant aerosol processes, the interactions between aerosol and radiation, and the interactions between aerosol and clouds (Stier et al., 2005; Zhang et al., 2012). It includes two options for the calculation of microphysics, the modal microphysics module M7 and the sectional microphysics module SALSA2.0 which was implemented in this study. The model design has been optimized for computational efficiency together with solving aerosol processes accurately. In its default setup, HAM uses the modal approach together with the model aerosol microphysics module M7 (Vignati et al., 2004). In this modal setup, the aerosol size distribution is described by a superposition of seven log-normal modes. Chemical components incorporated in each mode are chosen so that only those compounds which are relevant in the real atmosphere for each size range of each mode are included in those modes. The external mixing of aerosol is considered such that the soluble and insoluble compounds are emitted in separate parallel modes and as the insoluble modes are aged (i.e. soluble compounds accumulate on insoluble modes) insoluble modes are merged into the soluble modes. The chemical compounds in HAM can be considered as compound classes in the sense that they group certain types of aerosols to model compounds. These compounds are "sulfate" (SU), "organic aerosol" (OA), "sea salt" (SS), "black carbon" (BC), and "mineral dust" (DU). In practice, each individual model compound represents several individual compounds and especially OA represents hundreds of different organic compounds (Kanakidou et al., 2005). However, using lumped components is a fairly standardized practice in global aerosol models and the model components are usually the same in most models (Mann et al., 2014). The exception are organic compounds which are often separated e.g. based on their formation mechanims, i.e. primary and secondary organic aerosol (Tsigaridis et al., 2014).

Processes and properties related to the aerosol particles which are simulated by HAMMOZ are: emissions, dry deposition, wet deposition, sulfur chemistry, sedimentation, radiative properties, microphysical processes, and relative humidity in the

cloud-free part of the grid cells. HAMMOZ simulated aerosol are also coupled to the ECHAM-HAMMOZ cloud scheme and affect liquid cloud droplet formation and ice crystal formation (see Lohmann et al. (2007)).

In the default model configuration, all of these processes are calculated using the modal approach and the microphysics are calculated using the M7 module. Thus, the implementation of SALSA requires also the modification of HAM routines to follow the sectional representation and allow for consistent representation of these processes for modal and sectional approaches.

## 2.3 SALSA

The aerosol microphysical model SALSA is designed to be applicable to different scales of aerosol modeling starting from 0-dimensional simulations of laboratory or chamber experiments (Kokkola et al., 2014). It has also been implemented in the large eddy simulations (LES) model UCLALES (Tonttila et al., 2017) for 1-, 2-, 3-dimensional simulations. SALSA has also been implemented in the chemical transport model MATCH (Andersson et al., 2015) which in turn has been coupled to the regional climate model RCA4 (Thomas et al., 2015). This scalability and usage of one model across different scales allows for the easy parameterization of small scale aerosol processes up to the global scale. On the global scale, SALSA has previously been implemented in ECHAM5-HAM (Bergman et al., 2012). Here we present the configuration of SALSA which has been implemented in ECHAM-HAMMOZ and builds upon the implementation of SALSA in ECHAM5-HAM. For clarity, in this section, the ECHAM5-HAM implementation is called SALSA1 and the one implemented in ECHAM6-HAMMOZ is called SALSA2.0.

SALSA represents the aerosol size distribution using the sectional approach. The size distribution is divided into 10 size classes using volume ratio discretization (Jacobson, 2005). However, the width of the size classes vary over three size ranges: Subrange 1 for particles with diameters $D_p = 3\,\mathrm{nm} - 50\,\mathrm{nm}$, Subrange 2 for $D_p = 50\,\mathrm{nm} - 700\,\mathrm{nm}$, and Subrange 3 for $D_p = 700\,\mathrm{nm} - 10\,\mathrm{\mu m}$. This separation was done so that the size resolution is highest in the accumulation mode sizes which increases the accuracy of the cloud activation calculations. For each size class the tracer variables are: the number of particles and the concentration of individual species.

In SALSA1, Subrange 1 assumed internal mixing for all sizes, Subrange 2 included two externally mixed size classes (soluble and insoluble), and Subrange 3 included three externally mixed size classes (soluble, fresh insoluble and aged insoluble). In addition, the number of chemical compounds varied between the three size ranges. In SALSA2.0, the width of the size bins remains unchanged from SALSA1. However, subranges 2 and 3 are now treated as one so that the combined size range includes two externally mixed size classes; one where the insoluble compounds are emitted and one where the soluble compounds are emitted. These subregions are visualized in Figure 1. The change in how the chemical compounds are treated was first of all due to practical reasons. In SALSA1, the information of individual species was lost when the particles grew to sizes larger than 700 nm in diameter. This caused problems in studies where the information of individual species in all particle sizes was required (e.g. Kipling et al., 2016). Second, although in the troposphere microphysics have very little influence on the size of particles in the third subregion, when simulating volcanic eruptions or stratospheric solar radiation management, condensation can grow the largest particles. This caused the model to have problems in simulating the growth of particles in a volcano plume since the third region particles did not grow. This in turn resulted in underestimation of the effective radius of the volcano plume

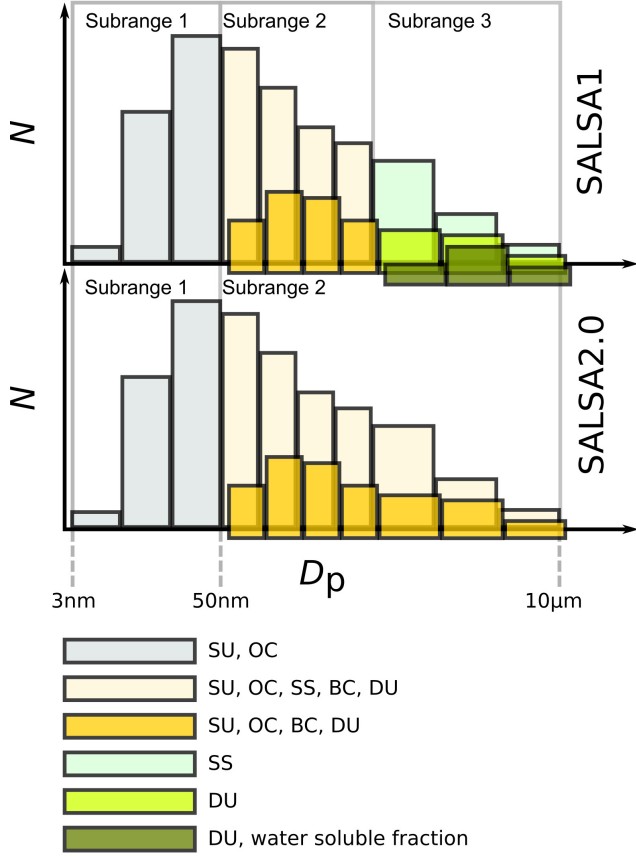

**Figure 1.** Schematic of the number ($N$) size distribution representation as a function of diameter $D_{\mathrm{p}}$ in SALSA1 (upper size distribution) and SALSA2.0 (lower size distribution). The color of each size class indicates which compounds are included in the size class.

(Kokkola et al., 2009). In 0-dimensional model tests (not shown here), SALSA2.0 did not exhibit such problems. Please note, that SALSA1 is no longer an optional aerosol microphysics module for the current or future releases of ECHAM-HAMMOZ.

Another significant change between SALSA1 and SALSA2.0 has been the modification of the aerosol size distribution update routine. In SALSA1, the moving center method (Jacobson and Turco, 1995) was used for Subranges 1 and 2, and the fixed sectional method (Gelbard et al., 1980) for Subrange 3. In SALSA2.0 the hybrid bin method (Young, 1974; Chen and Lamb, 1994) is used for all size sections. This is because the moving center method has been shown to introduce numerical artefacts in zero dimensional box model simulations (Mohs and Bowman, 2011) and when simulating aerosol formation and growth in high sulfur concentration conditions typical for e.g. large volcanic eruptions and simulations of stratospheric solar radiation management (e.g., Kokkola et al., 2008, Figure 2). In addition, in the study by Bergman et al. (2012), SALSA1 underestimated aerosol number concentrations observed at ground stations when using the moving center method. Switching to the hybrid bin method decreased the low bias.

**Table 1.** Overview of the treatment of different aerosol processes in the sectional approach (SALSA) and the modal approach (M7) when using the default setup.

| | SALSA2.0 | M7 |
|---|---|---|
| **microphysical process:** | | |
| nucleation | activation type nucleation (Sihto et al., 2006) | neutral and charged nucleation of $H_2SO_4$ and $H_2O$ (Kazil and Lovejoy, 2007) |
| condensation of $H_2SO_4$ | analytical predictor of condensation solved simultaneously with nucleation (Jacobson, 2005) | Two-step operator splitting scheme with an analytical solution for production and condensation (Kokkola et al., 2009) |
| coagulation | semi-implicit method (Jacobson and Turco, 1995) | implicit method (Vignati et al., 2004) |
| hydration | ZSR method (Stokes and Robinson, 1966) | $\kappa$-Köhler (Petters and Kreidenweis, 2007) |
| **emissions:** | | |
| Sea Salt | Size segregated sea salt emissions from Long et al. (2011) parameterization mapped to the soluble size sections in subrange 2 following the M7 mode parameters for accumulation and coarse modes. | Size segregated sea salt emissions from Long et al. (2011) parameterization mapped to the soluble accumulation and coarse modes |
| Mineral Dust | Size segregated mineral dust emissions from Cheng et al. (2008) parameterization mapped to the insoluble size sections in subrange 2 following the M7 mode parameters for accumulation and coarse modes. | Size segregated mineral dust emissions from Cheng et al. (2008) parameterization mapped to insoluble accumulation and coarse modes |
| **radiative effects** | Look-up-tables which are based on mie calculations for the extinction cross section, asymmetry factor, and single scattering albedo as a function of Mie size parameter and refractive index. Size sections are assumed to have a "flat top" size distribution within bins | Look-up-tables which are based on mie calculations for the extinction cross section, asymmetry factor, and single scattering albedo as a function of Mie size parameter and refractive index. Look-up-tables have been pre-calculated separately for modes with geometric standard deviation of 1.59 and 2.0 |
| **below-and in-cloud scavenging** | Prescribed scavenging coefficients for each size section according to Bergman et al. (2012) | Prescribed impaction scavenging coefficients for each mode according to (Stier et al., 2005) or size dependent scavenging rates according to (Croft et al., 2009, 2010) |

The implementation of SALSA2.0 in ECHAM-HAMMOZ was designed such that it shares the routines with the modal scheme of M7 wherever possible. In the sense of model processes, the biggest difference is in the aerosol microphysical calculations which are treated using methods that are designed for the respective size distribution description. The microphysical processes and other aerosol processes that are treated differently between the two model configurations are listed in Table 1. A comprehensive review of the relative importance of these processes within the ECHAM framework has been given previously by Schutgens and Stier (2014).

For offline emissions of SU, OA, and BC, SALSA2.0 uses the emission size distributions of M7 which are remapped to SALSA2.0 size sections. The details of these emission size distributions for different chemical compounds and emission sectors are given in Zhang et al. (2012) (Table 2).. Online emissions for SS, and DU are calculated online according to Long et al. (2011) and Cheng et al. (2008), respectively.

## 3 Model simulations

As the base simulation, we run ECHAM-HAMMOZ with SALSA2.0 for a ten year period (2003-2012) which was preceded by a one year spin-up period. The large scale meteorology (vorticity, divergence, and surface pressure) was nudged towards the ECMWF (European Centre for Medium-Range Weather Forecasts) reanalysis data ERA Interim (Berrisford et al., 2011). The relaxation times for the nudging of the surface pressure, vorticity, and divergence are $24\,\mathrm{h}$, $6\,\mathrm{h}$, and $48\,\mathrm{h}$, respectively. For sea surface temperatures and sea ice distributions we used the monthly mean climatologies from the Atmospheric Model Intercomparison Project (AMIP) of the Program for Climate Model Diagnosis & Intercomparison (PCMDI) http://www-pcmdi.llnl.gov/projects/amip/ (Taylor et al., 2012). The mass emission fluxes of each aerosol species from anthropogenic sources are based on Aerocom II - ACCMIP emissions (Lamarque et al., 2010) which for the period 2000-2100 have been linearly interpolated to the Representative Concentration Pathways (RCP) projection RCP4.5 (van Vuuren et al., 2011). For the mass emission fluxes of individual species from biomass burning we used the GFASv1 database multiplied by a factor of 3.4 following the recommendation by Kaiser et al. (2012). Emissions of OA from biogenic sources, were based on the Aerocom I monoterpene emissions (Dentener et al., 2006) of which $15\,\%$ was assigned to the particle phase OA mass. For the terrestrial emissions of dimethylsulfide (DMS) we used the Pham et al. (1995) emission dataset and the oceanic DMS emissions were calculated online according to Kloster et al. (2006).

The model output consisted of instantaneous values at $3\,\mathrm{h}$ interval. Although ECHAM-HAMMOZ includes the explicit chemistry model MOZ, it was not used in these simulations. Instead, we used the simplified sulfur chemistry scheme of HAM (Feichter et al., 1996; Zhang et al., 2012). The module calculates the oxidation of $DMS$ and $SO_2$ by OH, $H_2O_2$, $NO_2$, and $O_3$ in the gas and the aqueous phase. The oxidant concentrations are prescribed using monthly mean 3-dimensional fields from the MOZART chemistry model simulation (Horowitz et al., 2003).

In order to evaluate how the sectional approach performs against the modal approach within the same atmospheric model, we repeated the simulations for the year 2010 using M7 as the aerosol microphysical module with a setup as similar as possible. In the default setups of M7 and SALSA2.0, wet deposition and secondary organic aerosol (SOA) formation are the only processes

(in addition to the calculation of aerosol microphysics) that use different methods for solving the physics of the process. For the rest of the processes the difference is only in the numerical treatment. To minimize the differences between simulations done with the sectional and modal versions, the wet deposition scheme for M7 was changed to use the same prescribed wet scavenging coefficients as were used for SALSA2.0 (see Table 1). These coefficients have also been used in M7 in previous versions of ECHAM-HAMMOZ (Stier et al., 2005; Zhang et al., 2012). The implementation of the M7 scavenging coefficients for SALSA1 size sections has been presented by (Bergman et al., 2012). The reason for using the older approach is that the implementation of an improved wet scavenging scheme in SALSA2.0 is still under development. However, in order to compare the significance of microphysical processing and wet deposition on the modeled aerosol, we ran one additional simulation for the year 2010 with M7 using the more physically basedsize dependent scavenging rates (Croft et al., 2009), i.e. the default configuration of ECHAM-HAMMOZ. On the other hand, it should be noted that a comprehensive evaluation of the default version of ECHAM-HAMMOZ with M7 will be given in a separate paper by Tegen et al. (2018) and thus we do not do a full evaluation of it here.

In addition to the wet deposition scheme, we also turned off the SOA formation routine to keep the model configurations similar in the evalution. The SOA schemes are very different in their approach, as M7 assumes equilibrium partitioning for SOA while SALSA2.0 calculates SOA partitioning kinetically, solving size resolved condensation equations. The SOA scheme will be presented in detail by a companion paper by Kuhn et al. (in preparation). Instead of the detailed SOA schemes detailed in Table 1, for both SALSA2.0 and M7 we used the Aerocom I monoterpene emissions (Dentener et al., 2006) of which 15 % was irreversibly assigned to the particle phase OA mass.

As the sectional method requires more tracer variables for representing aerosol size dependence, SALSA2.0 is computationally slower that M7. The computation time depends very much on the time interval and the number of output variables. With Cray XC 30 architecture using 120 CPU cores, the evaluation simulations of SALSA2.0 took approximately double the time of M7.

## 3.1 Pinatubo experiment

Previous 2D (Herzog et al., 2004; Weisenstein et al., 2007) and box model (Kokkola et al., 2009) studies have shown that the modal approach, especially when the mode width is prescribed, can not reproduce aerosol growth when the concentration of condensing species is very high (Weisenstein et al., 2007; Kokkola et al., 2009). This can be the case in simulating stratospheric sulfur solar radiation management or in the case of strong volcanoes which emit high concentrations of sulfur into the stratosphere. Using the default mode width of M7 in high sulfur concentrations the growth of the aerosol effective radius is too rapid and leads excessive removal of stratospheric aerosol by sedimentation (Kokkola et al., 2009). This is because the high concentration of sulfur produces a bi-modal aerosol population seen in model simulations (Kokkola et al., 2009) and observations after the Mt Pinatubo eruption (Deshler et al., 2003; Deshler, 2008). The width of the aerosol size distribution is narrow because the smaller the particles are the faster they grow by condensation as the surface-to-volume ratio increases with decreasing particle size (Turco and Yu, 1999). Such size distributions were also observed after the Pinatubo eruption

(Deshler et al., 1997). If prescribed widths are used for the modes, the volume mean diameter, i.e. the diameter that dictates the sedimentation velocity of the modes, grows fast resulting in particles sedimenting faster (Kokkola et al., 2009).

An alternative approach for M7 in simulations of high statospheric sulfur load is to change the geometric standard deviation to 1.2 in the accumulation mode and remove the coarse mode. This modal setup has been shown to improve the ability of the model to reproduce the aerosol growth in high sulfur stratospheric conditions (Kokkola et al., 2009) and has been used in several studies related to stratospheric aerosol(Niemeier et al., 2009, 2011; Niemeier and Timmreck, 2015; Niemeier and Schmidt, 2017). However, we have to emphasize that such a setup is not a feature in the release version ECHAM6.3.0-HAM2.3-MOZ1.0 and using such a setup would require code level changes and obtaining suitable look-up-tables for the radiation calculations.

One commonly used test case (see e.g., English et al., 2013; Laakso et al., 2016; Timmreck et al., 2018) to evaluate how models perform in simulating high sulfur conditions is the eruption of Mt Pinatubo ($15.14°$ N, $120.35°$ E) in 1991. It has been estimated that the volcano emitted approximately 14 to $23\,\mathrm{Tg\,S}$ of $SO_2$ at $24\,\mathrm{km}$ altitude (Read et al., 1993; Guo et al., 2004). Here we used the mean of this range ($8.5\,\mathrm{Tg\,S}$). The oxidation of emitted $SO_2$ and the consequent new particle formation and growth of sulfate particles perturbed the stratospheric aerosol layer for over 3 years (Read et al., 1993; Guo et al., 2004).

To investigate how our model reproduces the aerosol properties of the Mt Pinatubo eruption, we ran three sets of transient (no nudging) simulation ensembles (5 ensemble members per set) using: SALSA2.0, M7, and M7 with 1.2 geometric standard deviation for the accumulation and coarse mode (denoted as M7strat). This modification applies also to the tropospheric aerosol. For each model configuration, the ensemble consisted of five 30 month simulations that were preceded by a one year spin up. In each ensemble run, we have perturbed offline anthropogenic aerosol emissions by values of the order of $10^{-6}$ which is an insignificant number for emission strengths but due to the chaotic nature of the atmospheric model, changes the model dynamics.

The emission settings are identical to those used by Niemeier et al. (2009) and Laakso et al. (2016). In addition, to see how much the current model differs from the previous generation model, we also included simulated aerosol properties from a MAECHAM5-SALSA simulation (Laakso et al., 2016) where the name MAECHAM5 refers to the middle-atmosphere configuration of ECHAM5. In this model setup, SALSA1 was modified so that subregion 2 was extended to cover subregion 3, similarly to SALSA2.0 in order to properly simulate the growth of the particles in high sulfur conditions (see Section 2.3).

## 4 Results

### 4.1 Aerosol optical properties

Satellite observations provide the best global coverage of aerosol optical properties and thus comparing the model with satellite retrievals gives a good indication of how the models perform in reproducing regional aerosol characteristics. Here we compared simulated aerosol optical depths (AOD) with those retrieved from the moderate-resolution imaging spectroradiometer (MODIS) instrument on board of both Aqua and Terra satellites (King et al., 1999).

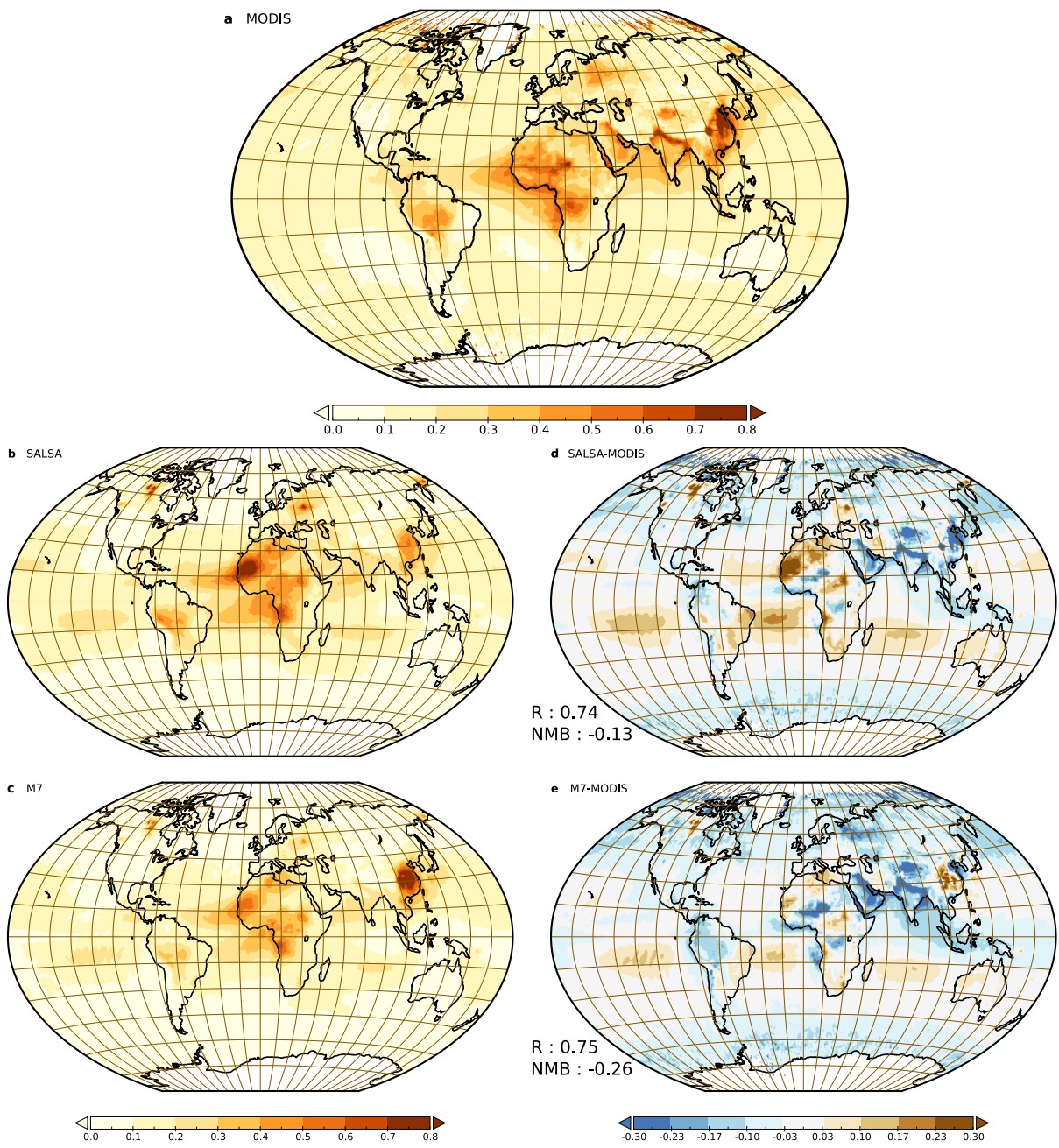

**Figure 2.** Yearly mean Aerosol Optical Depth for the year 2010 retrieved by a) MODIS (Aqua and Terra combined), and modelled by b) SALSA2.0 (model data collocated with Aqua and Terra retrievals), c) M7 (model data collocated with Aqua and Terra retrievals). Absolute differences between d) MODIS and SALSA2.0, e) MODIS and M7.

The ground based sun photometers also provide good coverage of observations of aerosol optical properties. Although they are column measurements covering a much smaller area than satellites, they are often considered as the "ground truth" of aerosol properties as they are less affected by the uncertain surface reflectance . Here we used the AOD retrievals from the sun photometer network AERONET (AErosol RObotic NETwork (Holben et al., 1998)) to evaluate the modeled aerosol optical properties.

### 4.1.1 Evaluation against MODIS observations

The model versus MODIS evaluation was made for the year 2010. From MODIS, we used the level 2.0 combined product of Deep Blue and Dark Target retrievals for $550\,nm$ wavelength aerosol optical depth (Sayer et al., 2014). It has been shown that, in order to get a representative comparison between model data and satellite observations, model data should be sampled at the time and the location of the satellite observations they are compared to (Schutgens et al., 2016). For this purpose, we used the Community Intercomparison Suite (CIS) tool (Watson-Parris et al., 2016) which was applied to collocate the model AOD with the observations.

From the figure, we can see that the overall comparison between both models and satellite data is generally good. For the yearly mean values, the correlation coefficient R between MODIS AOD and SALSA AOD is 0.74 and for M7 it is 0.75. The normalized mean bias (NMB) for SALSA is -0.13 while for M7 it is -0.26. Areas that exhibit the largest differences between the models and observations are 1) the Saharas, 2) highly polluted areas over India and southeast Asia, 3) regions with high AOD due to biomass burning over Russia, Canada, Central Africa and South America. These are regions which are strongly affected by primary emissions. However, over these areas also SALSA2.0 and M7 have noticeable differences in the simulated AODs which means that the aerosol representation has a significant effect on the modelled AOD. Over the Sahara, the most significant contribution to the AOD comes from mineral dust. Since dust emissions in ECHAM-HAMMOZ are very sensitive to small changes in 10-meter wind speed, changes in the wind speed can cause large changes in dust emissions even if the model meteorology is nudged (Bergman et al., 2012). This is because the nudging does not strictly force the model meteorology to reanalysis data. Consequently, difference in the model dynamics which result in changes in DU emissions explains the difference in AOD values between the two model configurations especially in the northwest regions of Africa where DU mass emissions in some of the grid boxes are more than 3 times higher in SALSA2.0 than in M7. This can be seen in Figure 3 which shows the relative change between SALSA and M7 mass emission strengths for DU.

However, over southeast Asia and biomass burning regions, simulated aerosol load is mostly dictated by offline emissions which are, in mass, identical for both model setups. Thus, differences over these areas predominantly come from the differences in the representation of the size distribution, the microphysical processing of aerosols, and sink processes. This can be seen when comparing the simulated composition and extinction distributions at two sites where the simulated AOD is mainly driven by aerosol compounds from offline emissions but where the AOD in SALSA2.0 significantly differs from those in MODIS and M7. Figure 4 shows the 2010 yearly mean mass and extinction size distributions for SALSA2.0 and M7 over China at a location (30.775° N, 114.375° E) where the simulated AODs are extremely high (Fig. 4a) and Russia at a location (55.025° N, 39.375° E) where biomass burning emissions are high (Fig. 4b). To make the visual comparison easier, the M7 size distributions

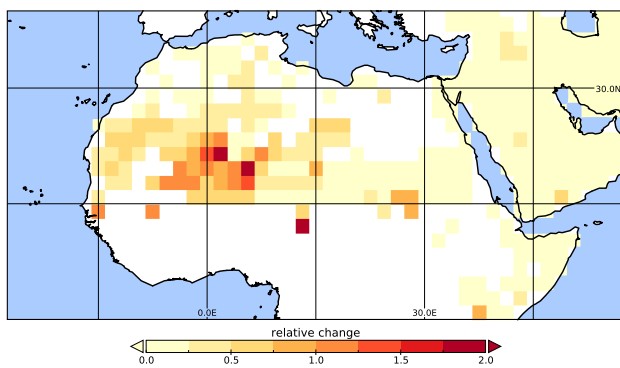

**Figure 3.** Relative change in the simulated yearly mean mass emission strenghts of DU between SALSA2.0 and M7. Grid boxes marked in white and blue are land and water grid boxes with no dust emissions in the model.

were remapped to SALSA2.0 size classes. At the Chinese site, AODs from SALSA, MODIS and M7 are 0.47, 0.87, and 1.13, respectively. At the Russian site, AODs from SALSA, MODIS and M7 are are 0.70, 0.42, and 0.44, respectively.

When analyzing the aerosol mass size distributions, it is evident that over these locations the aerosol extinction is strongly affected by the differences between SALSA2.0 and M7 in the methods used for calculating microphysical processes, espe-
cially gas-to-particle partitioning. For calculating concurrent nucleation and condensation, M7 uses the method introduced by Kokkola et al. (2009) and SALSA2.0 the method by Jacobson (2005). In the upper panels of Figure 4 we show the mass size distribution for SALSA2.0 and M7. In each class, the mass fraction of each compound is indicated by a color. Figure 4 shows that the largest difference in the composition distribution comes from SU, which is the only condensable species in this model configuration. Compared to M7, SU in SALSA2.0 is more evenly spread among all sizes, and there is a relatively higher
amount of sulfate in the largest sizes. This difference is very likely due to the numerical limitations of the modal scheme. The modal scheme has been shown to overestimate the condensational growth of the accumulation mode thus underestimating the amount of condensable species in the largest particles (Zhang et al., 1999). In addition, in the modal approach the mass distribution of all compounds follow the shape of the mode restricting the mass distribution of individual compounds. It has to also be noted that the emission size distributions are not optimal for M7 as the emissions in each mode are assumed to have a
fixed radius. The same applies to SALSA2.0 since the emission size distribution assumed the same shape as M7.

The extinction at $550\,\mathrm{nm}$ wavelength for different sized particles at $70\,\%$ relative humidity are shown in the lower panel of Figure 4. The aerosol extinction is a quantity which is highly nonlinearly dependent on the aerosol size, aerosol hygroscopicity, and relative humidity. Thus although the differences in SALSA2.0 and M7 simulated aerosol are caused by similar processes, differences in the simulated extinctions can switch signs at different sites. At the Chinese site the resulting shape of the size
distribution of M7 yields a higher aerosol extinction than SALSA2.0 while at the Russian site, it is the opposite. At the Russian site the composition distributions of both OA and SU are significantly different between the two model versions. This is because

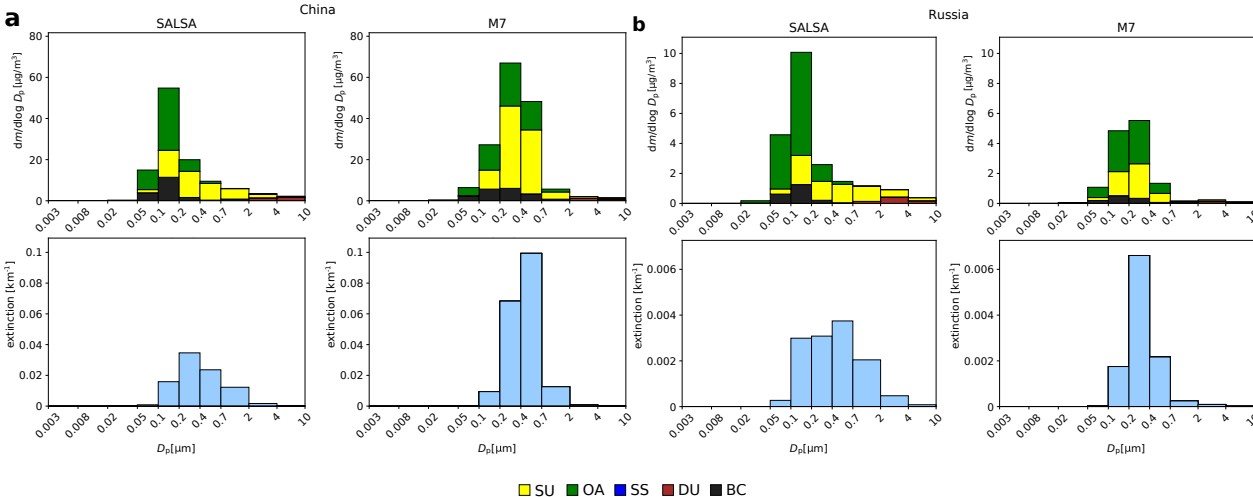

**Figure 4.** SALSA2.0 and M7 simulated mass ($\mathrm{d}n/\mathrm{dlog}D_\mathrm{p}$) and extinction size distribution in a) China ($114.375°$ E, $30.775°$ N) and b) Russia ($39.375°$ E, $55.025°$ N). The height of the bars in the upper row represents the number concentration of particles $\mathrm{d}N/\mathrm{dlog}D_p$. The colorbars represent the mass fraction of each chemical compound in each size class. In the bottom row the height of the bars denote the extinction of the size classes.

OA is not included in the insoluble accumulation mode in M7 while in SALSA2.0, both soluble and insoluble size classes include OA. Wet removal is faster for soluble particles which results in faster removal of OA accumulation sized particles in M7. The overestimation of AOD with both model setups at the Russian site indicates that biomass burning emissions are overestimated.

It should be noted that over China MODIS has been shown to have a high bias in AOD when compared to AERONET observations (Lipponen et al., 2017). Especially over the highly polluted areas in China this high bias is likely to increase the discrepancy between the SALSA2.0 simulated AODs and MODIS AODs.

Figure 5 shows the zonal mean AOD for MODIS together with SALSA2.0 and M7 model data. To visualize how the wet deposition scheme affects the zonal AOD we also included the zonal mean AOD from the simulation with the aerosol size-
dependent wet deposition scheme, which is used as the default scheme in ECHAM-HAMMOZ (denoted M7default in Fig. 5).

As can be seen from the figure, the modelled AOD decreases faster when moving from the equator towards the poles in comparison to the satellite observations. This is the case for both M7 and SALSA2.0 and has been apparent also in previous model versions (Stier et al., 2005; Bergman et al., 2012). Compared to the previous model versions, the decrease in AOD
towards the South Pole has been further amplified due to the new Long et al. (2011) sea salt emission parameterization. This is because sea salt mass emissions decrease significantly in ECHAM-HAMMOZ when using Long et al. (2011) in comparison to the previously used Guelle et al. (2001) implementation.

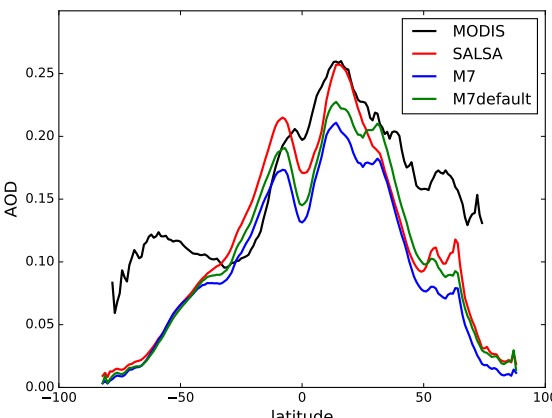

**Figure 5.** Zonal mean of aerosol optical depth in year 2010 observed by MODIS (Aqua and Terra combined), SALSA2.0 (red curve), M7 (blue curve), and M7 with default wet deposition scheme (green). AOD values from MODIS at high latitudes were excluded due to the larger retrieval uncertainty at high latitudes.

Overall, the zonal average of SALSA2.0 is in a better agreement with the observations than M7 except between the latitudes 10°S -35°S. Over these latitudes, the AOD is overestimated compared to MODIS. This is caused by biomass burning aerosol for which the emissions are likely overestimated. Similar to biomass burning regions in Russia, SALSA2.0 produces higher AOD than M7 over biomass burning influenced regions over Africa and South America affecting also AOD over the oceans in this latitude band 10°S -35°S.

Over the Northern Hemisphere, the magnitude of the zonal gradient of AOD in ECHAM-HAMMOZ is strongly dependent on the wet deposition scheme (Bourgeois and Bey, 2011). From Figure 5, it can be seen that compared to M7 the improved wet deposition scheme (M7default) increases the AOD towards the Arctic improving the comparison between the model and MODIS. The improved wet deposition scheme affects the AOD gradient to a similar degree as the choice of the aerosol microphysics scheme. For example, SALSA2.0 and M7default AOD values overlap in the sub-Artic and the Arctic region and, on average, the difference between M7 and M7default is smaller than the difference between M7 and SALSA.

The global mean AOD is also underestimated with both model setups although the bias in SALSA2.0 is smaller than in either of the M7 setups. Especially, the tropical maximum is better captured with SALSA. The observed global mean AOD from MODIS (Aqua and Terra combined) is 0.170 while the modelled values for MODIS collocated AOD are for SALSA2.0: 0.145, for M7 0.122, and for M7default 0.136. Figure 5 indicates that the low bias near in the high latitudes can partly be explained by low SS emissions, especially in the Southern Hemisphere. On the other hand, it has been previously shown that insufficient aerosol transfer in ECHAM-HAMMOZ can also partly explain low aerosol mass over the high latitudes (Bourgeois and Bey, 2011; Kristiansen et al., 2016). In addition, it has to be noted that except for South Africa and Oceania, MODIS overestimates AOD compared to AERONET observations (Lipponen et al., 2017).

### 4.1.2 Evaluation against AERONET observations

For comparing the model data with the AERONET sun photometer observations, we used the whole simulated (2003-2013) period for SALSA2.0 and the simulated year 2010 for M7. The level 2.0 daily AOD data from AERONET was collected for all available 984 stations. Simulated daily means were sampled for the days where AERONET observations are available and they were also spatially collocated to the location of the AERONET station. Afterwards, a yearly average of both observed and simulated daily means were computed.

Figure 6 shows the scatterplots of SALSA2.0 modelled AOD against AERONET observed AOD. Fig 6a illustrates that the model AOD correlates well with the observations for the years 2003 – 2012. This is also reflected in the statistical values of the comparison as the correlation coefficient R is 0.79 and the normalized mean bias is -0.09.

In the year 2010 comparison (see Fig 6c), the correlation coefficient decreases slightly to 0.73 and the NMB reduces to a value -0.03. M7 (see Fig 6d) shows also a very good correlation with the AERONET observations with correlation coefficient of 0.71 and bias of -0.05.

In Fig 6, different regions are separated by color. From this separation, we can see that although statistical values are comparable between M7 and SALSA2.0, similar to the comparison with MODIS, there are regional differences. Regional AOD values together with their correlation coefficient values are listed in Table 2. From these values, we can see that AOD in both model setups is biased low compared to AERONET AOD. For example, SALSA2.0 underestimates AOD in 7 out of 10 regions. However, the correlation coefficient values are high for both models. The exceptions are Europe and Asia where the correlation coefficient values R are 0.57 or less for both SALSA2.0 and M7. In 3 out of 10 regions, SALSA has a higher correlation coefficient than M7 while the number of regions where each model has a lower bias is evenly divided. Asia is the only region where the AOD in M7 is higher than in SALSA.. Over Asia, SALSA significantly underestimates the AOD (shown by dark red markers in Figure 6) which was also the case in the evaluation against MODIS data. As was shown in Section 4.1.1 the treatment of microphysical processes, especially gas-to-particle partitioning, can significantly affect the number and composition of aerosol over highly polluted regions causing differences in the modeled AOD between the sectional and modal setups. However, the differences between the simulated and AERONET AOD are not as evident as in the MODIS evaluation. One reason for this is that MODIS AOD is biased high over Asia, especially over highly populated regions of China (Lipponen et al., 2017).

### 4.2 Aerosol mass concentrations at the surface

To evaluate the simulated aerosol mass concentrations at the surface, we compared the model data with those measured by the European Monitoring and Evaluation programme (EMEP; http://www.emep.int) and the United States Interagency Monitoring of Protected Visual Environment (IMPROVE; http://vista.cira.colostate.edu/improve/). Both of these observation networks provide data for the mass concentrations of individual chemical components of the aerosol and the data are freely available from both sources. From the European EMEP programme and the USA based IMPROVE monitoring sites we used the PM2.5 aerosol mass concentration data for sulfate and elemental carbon. Additionally, from IMPROVE we used the data for organic

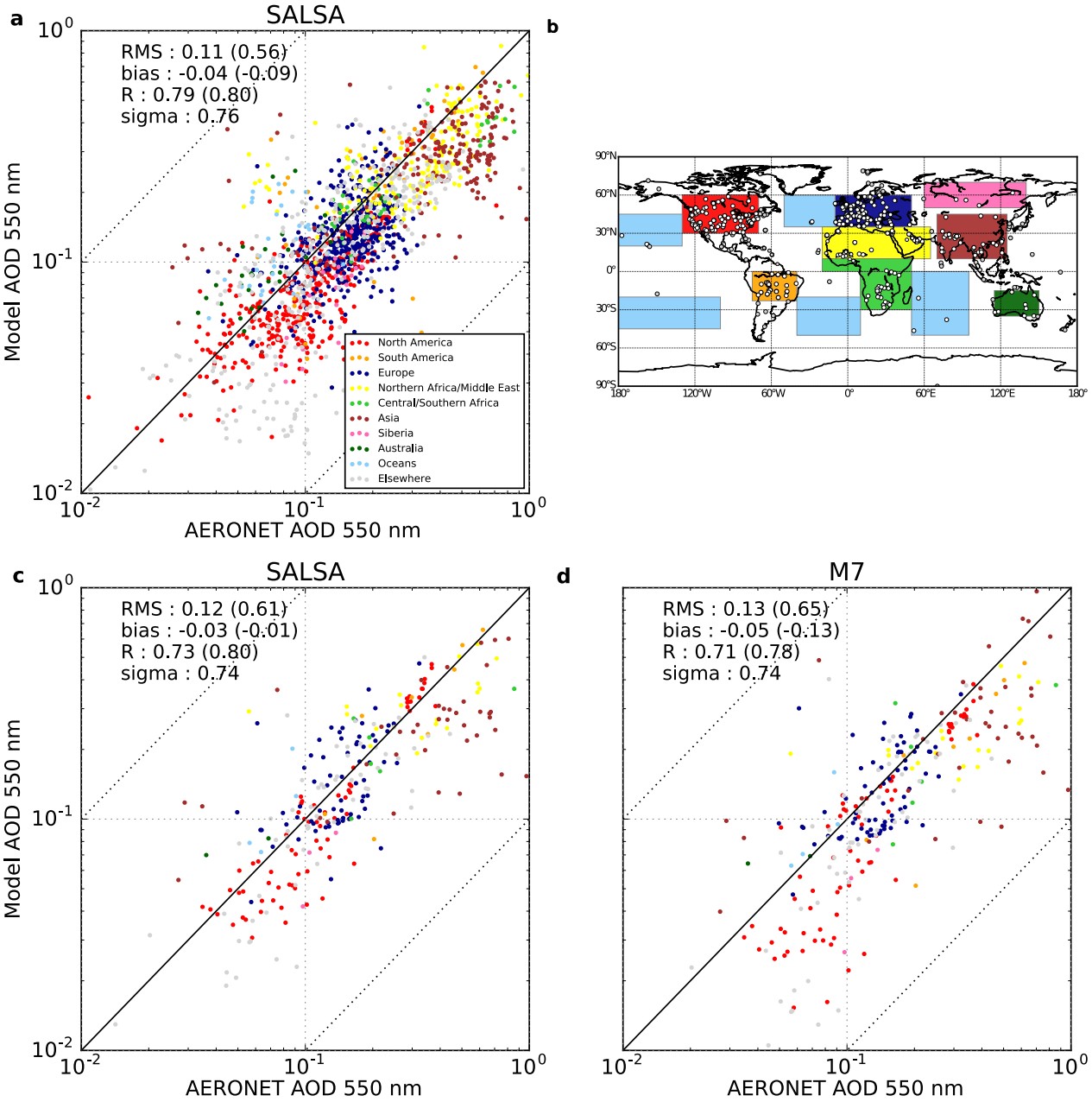

**Figure 6.** Scatterplots of yearly mean of daily AERONET AOD value against yearly mean of collocated simulated daily mean AOD. Panel a) represents the comparison between AERONET and SALSA2.0 for the period 2003-2012. Colors in the scatterplots denote different regions shown in the map in Panel b). Panel c) shows the comparison between AERONET and SALSA2.0 for the year 2010, d) comparison between AERONET and M7 for the year 2010. The given statistical values are the following: Root Mean Square RMS (normalised RMS), absolute bias (normalised bias), correlation coefficient R (R on log scale) and the ratio between simulated and observed standard deviation (sigma).

**Table 2.** Yearly means of daily AOD values from AERONET, SALSA, and M7 and the corresponding correlation coefficient values for the models.

| Region | AERONET | SALSA | | M7 | |
|---|---|---|---|---|---|
| | AOD | AOD | R | AOD | R |
| North America | 0.143 | 0.135 | 0.97 | 0.111 | 0.95 |
| South America | 0.343 | 0.338 | 0.92 | 0.246 | 0.95 |
| Europe | 0.149 | 0.157 | 0.57 | 0.143 | 0.53 |
| Northern Africa/Middle East | 0.366 | 0.321 | 0.66 | 0.239 | 0.69 |
| Central/Southern Africa | 0.298 | 0.216 | 0.78 | 0.207 | 0.70 |
| Asia | 0.427 | 0.264 | 0.47 | 0.286 | 0.41 |
| Siberia | 0.113 | 0.067 | 0.86 | 0.052 | 0.88 |
| Australia | 0.052 | 0.076 | 1.00 | 0.067 | 1.00 |
| Oceans | 0.075 | 0.120 | 0.79 | 0.097 | 0.75 |
| Elsewhere | 0.139 | 0.127 | 0.73 | 0.113 | 0.69 |

carbon. In total, data from 530 stations were used in the comparison. The comparison between SALSA2.0 and the surface observations was done for the period 2003-2012. From the model, we used the daily mean data sampled according to the days when there were observations at each station. To evaluate the difference between SALSA and M7, we also compared the simulated data for mass concentrations of SU, BC, and OA for the year 2010 against EMEP and IMPROVE observations.

In order to evaluate the simulated DU and SS mass concentrations, i.e. compounds whose emissions are wind driven, we compared the simulated masses against two sets of observations. Simulated dust masses were compared with the observations which were used in the Aerocom experiment by Huneeus et al. (2011) where 15 global models were compared to observations related to desert dust aerosols. Surface mass concentrations of DU were provided for the Pacific Ocean sites from the sea/air exchange SEAREX program (Prospero et al., 1989) and for the northern Atlantic sites from the Atmosphere-Ocean chemistry

experiment AEROCE (Arimoto et al., 1995). The AEROCE observations include also data for SS surface mass concentrations which were used in evaluating the simulated SS mass concentrations.

### 4.2.1 Sulfate

Figure 7 shows the scatterplot of observed and modelled yearly mean PM2.5 concentrations of SU. The left column of the figure shows the data for EMEP stations and the right column shows the data for IMPROVE stations.

Similar to the comparison to the AERONET AOD, SU mass concentrations from SALSA2.0 simulations correlate well with the observed surface concentrations. The correlation coefficient for SU for EMEP sites is 0.72 and for IMPROVE sites it is 0.89. SALSA2.0 tends to overestimate SU for both EMEP (NMB of 0.25) and IMPROVE (NMB of 0.33) stations. The high bias of aerosol mass concentration of SU over the US is in contrast to the underestimation of AOD by the model in these regions when

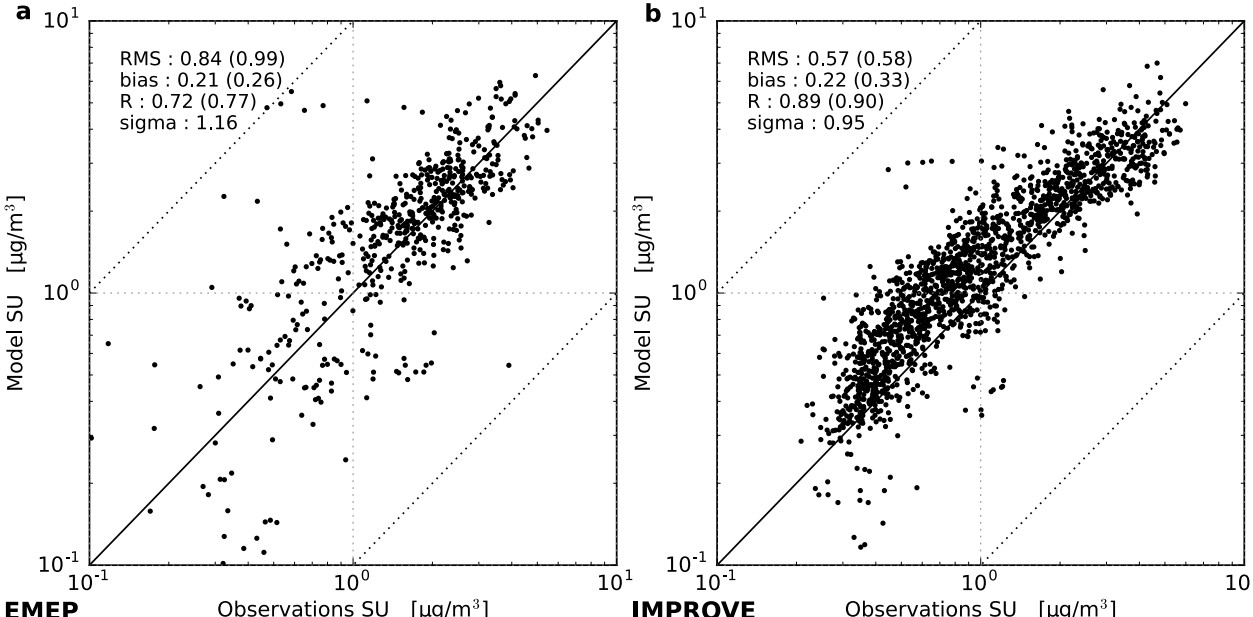

**Figure 7.** Scatterplots of yearly mean aerosol mass concentrations observed at EMEP (left column) and IMPROVE (right column) stations versus those from SALSA2.0 simulations for SU. The given statistical values are the same as in Figure 6

compared to MODIS and AERONET AOD. This highlights the sensitivity of AOD to the shape of the aerosol size distribution. Aerosol water has also a significant contribution to AOD and simulated relative humidity and aerosol hygroscopicity can cause differences between the simulated and observed AOD. In addition, in these regions, nitrate is a significant source of aerosol mass (Bauer et al., 2007) and as it is missing in our model it may also be a cause for the differences between model and observations, although the representation of nitrate in coarse resolution models is not without complications (Weigum et al., 2016).

The evaluation was repeated for the year 2010 in orded to include M7 in the comparison. For this simulation year, the correlation coefficient values for SALSA2.0 simulated SU mass concentrations were 0.60 for EMEP stations and 0.93 for IMPROVE stations. SALSA2.0 simulated SU mass concentrations in year 2010 had higher positive bias than those for the whole simulation period for both EMEP (NMB of 0.40) and IMPROVE (NMB of 0.42). For M7, the corresponding correlation coefficient values were 0.62 for EMEP and 0.93 for IMPROVE stations. Although, M7 had also high biases for EMEP stations (NMB of 0.23) and IMPROVE stations (NMB of 0.27), they were lower than for SALSA2.0.

### 4.2.2 Black Carbon

Figure 8 shows the scatterplots of observed and modelleing yearly mean PM2.5 concentrations of BC for the whole simulation period. The left column of the figure shows the comparison for EMEP stations and the right column shows the comparison for IMPROVE stations.

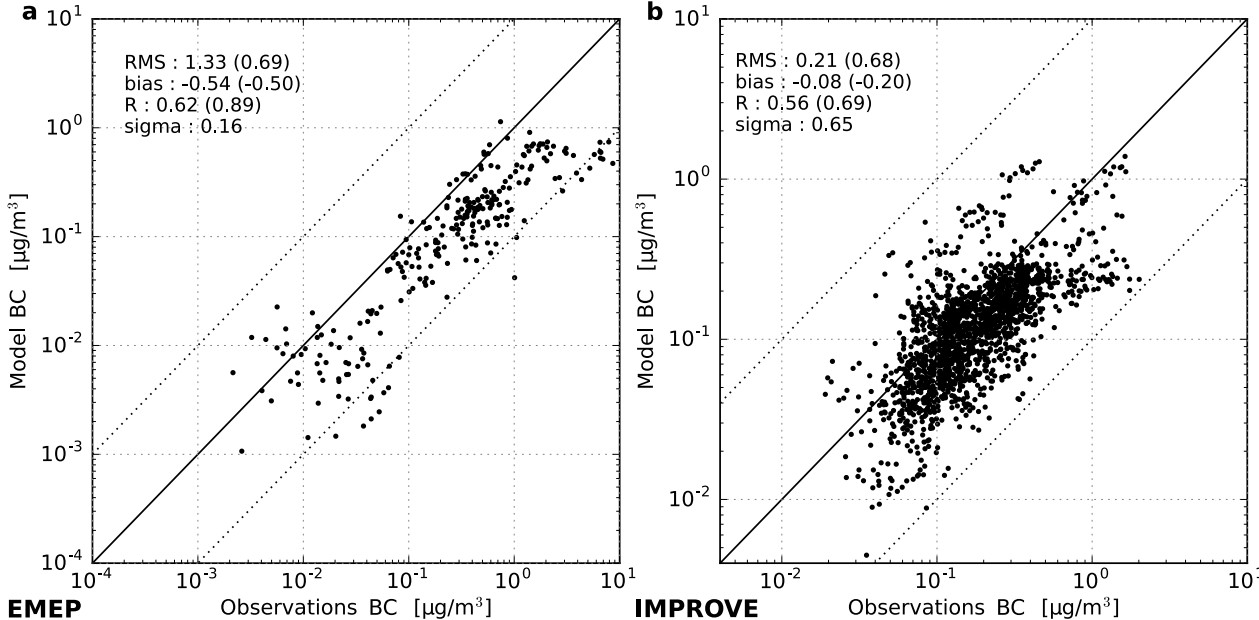

**Figure 8.** Scatterplots of yearly mean aerosol mass concentrations observed at EMEP (left column) and IMPROVE (right column) stations versus those from SALSA2.0 simulations for BC for years 2003-2012. The given statistical values are the same as in Figure 6

Compared to sulfate, for BC, the correlation is slightly lower with the correlation coefficient being 0.62 for EMEP and 0.56 for IMPROVE sites. In contrast to SU, BC mass concentrations are underestimated for both EMEP (NMB of -0.50) and IMPROVE (NMB of -0.20).

Similar to the evaluation of SU mass concentrations, the evaluation was repeated for the year 2010 including M7 in the comparison. For this simulation year, the correlation coefficient values for SALSA2.0 simulated BC mass concentrations were 0.42 for EMEP stations and 0.65 for IMPROVE stations. SALSA2.0 simulated mass concentrations in year 2010 were biased low, similarly to the whole simulation period, for both EMEP (NMB of -0.48) and IMPROVE (NMB of -0.21). For M7, the corresponding correlation coefficient values were 0.34 for EMEP and 0.64 for IMPROVE stations. M7 was also biased low for both EMEP stations (NMB of -0.53) and IMPROVE stations (NMB of -0.31).

### 4.2.3 Organic Aerosol

Surface mass concentrations of OA were compared to the IMPROVE observations. The data was available only for years until 2004 so here we compared the simulated year 2010 to observations for the year 2004 in order to get a better comparison between SALSA2.0 and M7. Figure 9 shows the scatterplots of observed OA surface mass against simulated values. Both models are biased low with NMB values of -0.56 and -0.59 and correlation coefficients of 0.40 and 0.42 for SALSA2.0 and M7, respectively. A more detailed evaluation of organic carbon will be carried out in a companion paper by Kuhn et al. (in preparation).

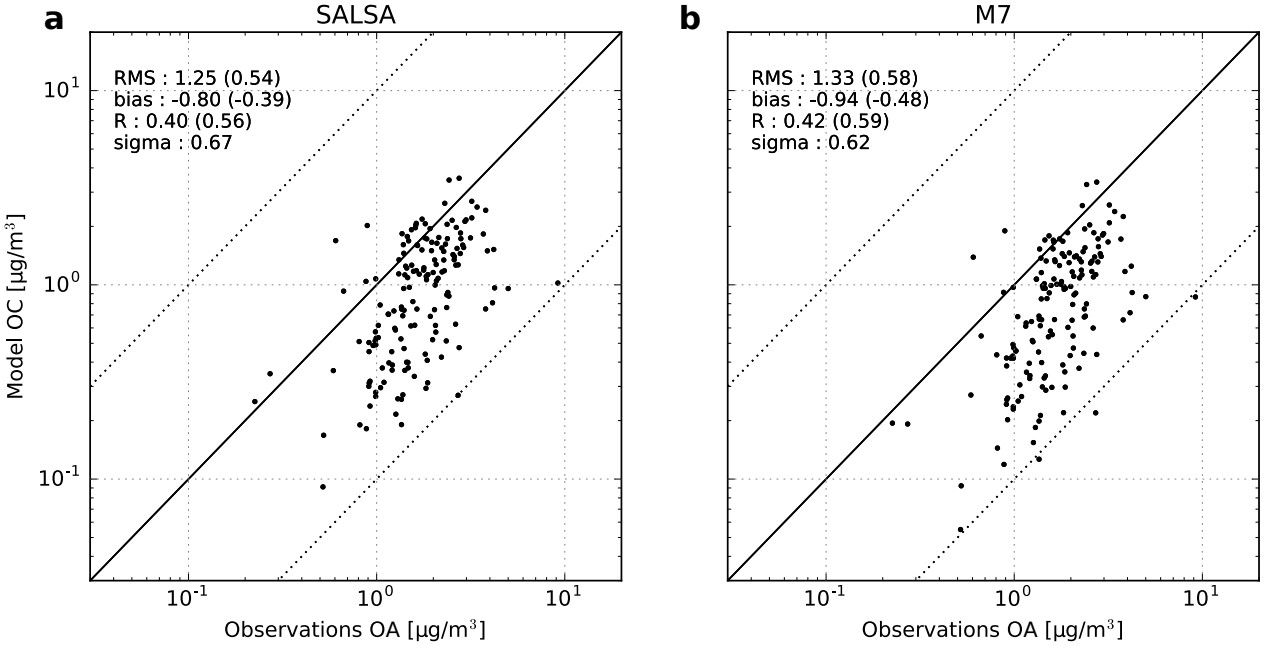

**Figure 9.** Scatterplots of yearly mean OA aerosol mass concentrations simulated and observed at IMPROVE sites (year 2004) for a) SALSA2.0 (simulated year 2010) and b) M7 (simulated year 2010). The given statistical values are the same as in Figure 6.

### 4.2.4  Mineral Dust

SEAREX was a 10-year (1977-1986) program and AEROCE a 5-year (1990-1995) program and thus outside of our simulation period, we compared the simulated data for the year 2010 to DU climatologies. The monthly model values were constructed by averaging daily means only for days where an observation is available. Moreover, each model monthly mean was spatially collocated to the location of the observation station (by bi-linear interpolation).

Figure 10 shows the scatterplots of monthly mean observed DU surface concentrations against those simulated using SALSA2.0 and M7. DU mass concentrations from both SALSA2.0 and M7 show a moderate agreement against observations but underestimate the low values. The correlation coefficients for SALSA2.0 and M7 are 0.66 and 0.47, respectively. Both SALSA2.0 and M7 exhibit low NMB with values of -0.33 and -0.26, respectively. It has to be noted that, due to different periods in observations and simulations, DU mass concentrations are not strictly comparable because they are very sensitive to the 10 meter wind speed.

### 4.2.5  Sea Salt

For evaluating mass concentrations SS we also used the data from SEAREX and AEROCE programs which were compared to the simulated SS mass concentrations for the year 2010. Figure 11 shows the scatterplots of observed monthly mean SS surface mass concentrations against simulated monthly mean surface mass concentrations from SALSA2.0 and M7. Collocation of the

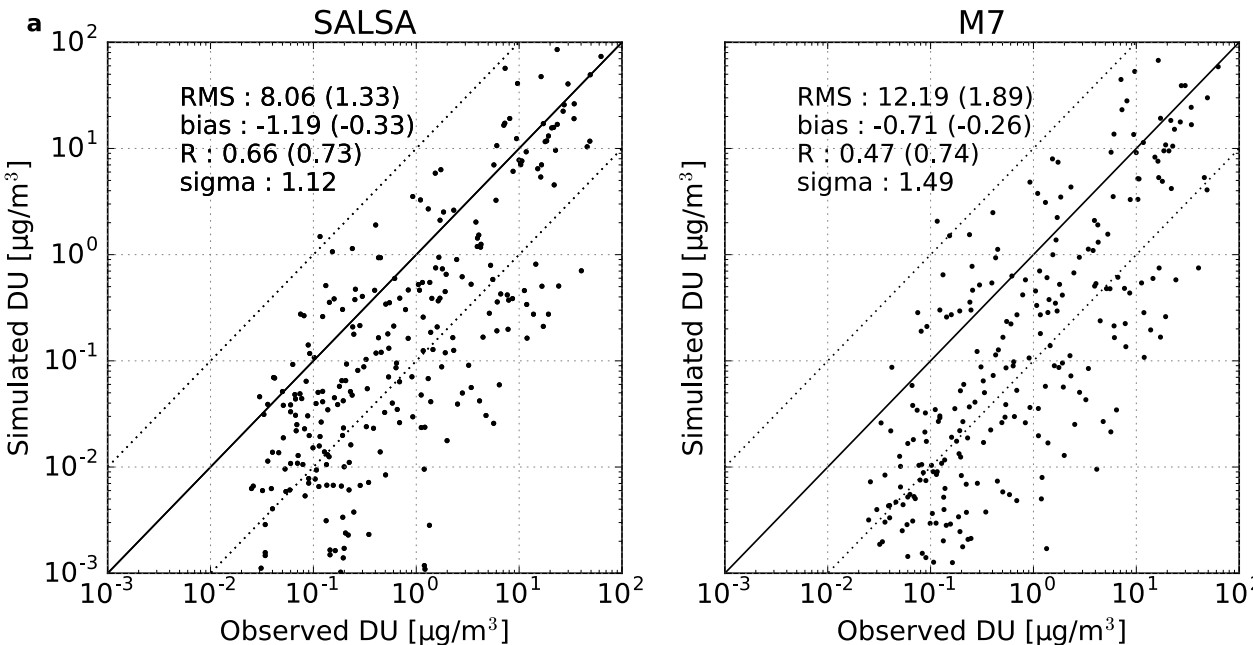

**Figure 10.** Scatterplots of aerosol masses observed in SEAREX program (years 1977-1986) and AEROCE (years 1990-1995) experiment against those from SALSA2.0 (year 2010) simulated aerosol masses for DU. The given statistical values are the same as in Figure 6

data was done identically to DU evaluation, described in the previous subsection. As was seen in the comparison between the models and MODIS retrievals, aerosol load over oceans south of latitude 40°S seems to be low in both model versions. This is also reflected in low SS mass concentrations in simulations when compared to the observations; in very few cases the values exceed the observed values. This indicates that the sea salt emissions are significantly underestimated in this model setup. The

5 NMB for SALSA2.0 and M7 were -0.68 and -0.64 while the correlation coefficients were 0.19 and 0.18, respectively. This may also explain the discrepancies between the model and satellite AODs over the oceans as sea salt strongly affects the aerosol size distribution over the oceans.

Since DU and SS emissions are calculated online, they vary annually. In order to evaluate, how much the choice of the year affects these results, we repeated the analysis for DU and SS for each year using the 10 year SALSA2.0 simulation. This

analysis showed that the main characteristics in the comparison between modeled and observed mass concentrations remain similar each year, i.e. the model has low bias in both DU and SS mass concentrations and the low model bias increases with decreasing mass concentration (for both DU and SS). For DU, the annual variability of the modeled mass concentration is fairly large with NMB ranging between -0.35 and -0.09. For SS the variability is low and the NMB varies between -0.74 and -0.70. The correlation between modeled and observed mass concentrations varies very little annually. For DU, the logarithmic

scale correlation coefficient varies between 0.67 – 0.74 for DU and 0.58 – 0.67 for SS.

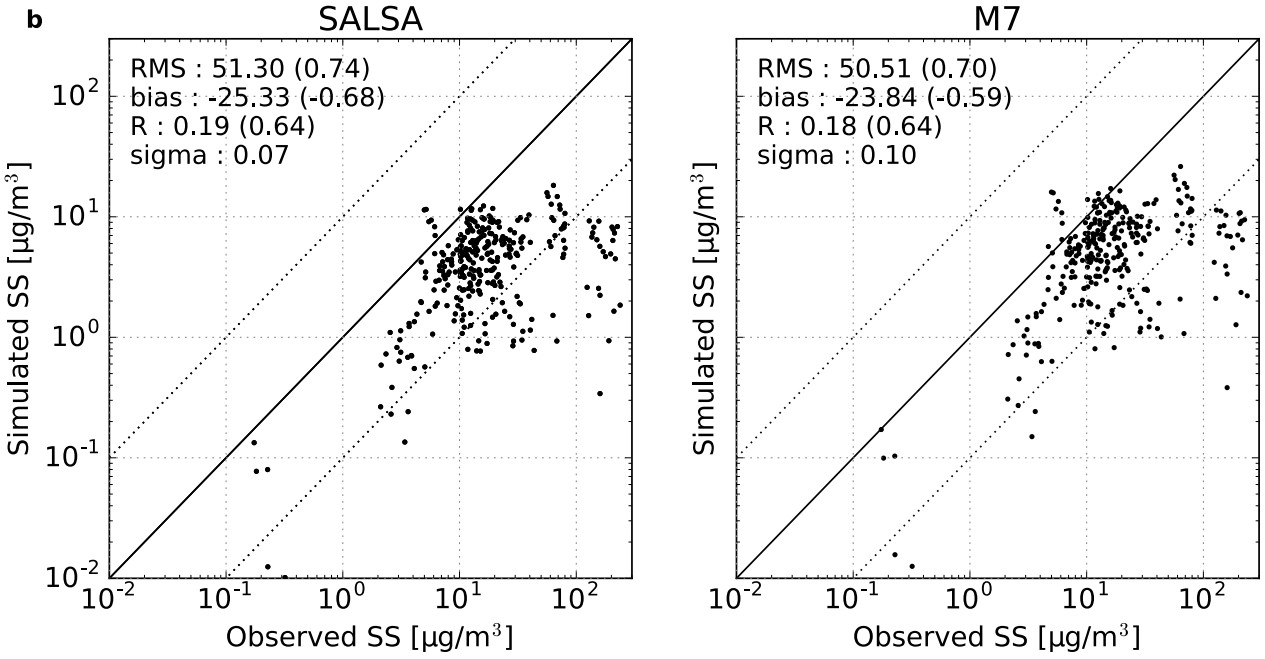

**Figure 11.** Scatterplots of aerosol masses observed in SEAREX program (years 1977-1986) and AEROCE experiment (years 1990-1995) against those from SALSA2.0 simulated aerosol masses for SS. The given statistical values are the same as in Figure 6

In addition, SS measurements are mostly at coastal sites where global models may have large biases in sea salt surface concentrations as SS emission parameterizations assume open ocean conditions (Spada et al., 2015). It has been suggested that caution should be taken when evaluating global models against coastal observations (Spada et al., 2015).

### 4.2.6 Summary

Table 3 summarizes the biases of simulated surface mass concentrations of SALSA2.0 and M7. In addition, as a reference, the table shows the same values from the previous model version ECHAM5-HAM-SALSA1 for the year 2008 which used the emissions for the year 2000 (Bergman et al., 2012). The table also shows the global burdens for these compounds for the same three model versions together with values reported by Liu et al. (2005) and Textor et al. (2006). Liu et al. (2005) has made a synthesis of model data and Textor et al. (2006) provides the analysis of global aerosol properties in Aerocom Phase I models

for the year 2000.

     From the table we can see that surface concentrations of sulfate and its global burden are significantly larger in SALSA2.0 than in the previous generation model and they are at the upper end of the estimate of (Liu et al., 2005). Although our simulation period is not for the same period as for ECHAM5-HAM, by Liu et al. (2005), and Textor et al. (2006), global sulfate emissions have been suggested to be fairly constant through 2000-2010 (Granier et al., 2011). Even larger increases between the two

model generations are evident for the BC and OA burdens which are approximately 3 times higher in ECHAM-HAMMOZ-

SALSA2.0. Despite these higher burdens the simulated BC and OA surface concentrations are biased low when compared to the observations from the IMPROVE network (see Figures 7, 8, and 9). The largest decrease in the burden can be seen for SS, which in SALSA2.0 has decreased to approximately 1/3 of the SS burden in ECHAM5-HAM supporting the conclusions of too low sea salt emissions in this model configuration. The DU burden has slightly increased between the two model generations with the DU burden being near the values of the Aerocom I mean.

**Table 3.** Comparison of mean NMB in ECHAM5-HAM (with SALSA1), ECHAM-HAMMOZ (with SALSA2.0), and ECHAM HAMMOZ (with M7) for individual compounds at IMPROVE sites, the global burdens (Tg) of all compounds together with those reported by Liu et al. (2005) and the mean of Aerocom I models analyzed by Textor et al. (2006).

| | ECHAM5-HAM-SALSA1 | ECHAM-HAMMOZ-SALSA2.0 | ECHAM-HAMMOZ-M7 | | |
|---|---|---|---|---|---|
| SU | 0.19 | 0.49 | 0.33 | | |
| BC | -0.24 | -0.21 | -0.31 | | |
| OA | 0.25 | -0.37 | -0.47 | | |
| Global burden (Tg) | | | | (Liu et al., 2005) | Aerocom I |
| SU (Tg S) | 0.64 | 0.96 | 0.74 | 0.53-1.07 | 0.66 |
| BC | 0.07 | 0.26 | 0.20 | 0.12-0.29 | 0.24 |
| OA | 0.96 | 2.68 | 1.77 | 0.95-1.8 | 1.70 |
| SS | 11.73 | 3.53 | 4.21 | 3.41-12.0 | 7.52 |
| DU | 13.11 | 18.26 | 15.14 | 4.3-35.9 | 19.20 |

## 4.3 Evaluation against aircraft observations

The previous evaluations showed how well the model reproduces surface concentrations and column quantities of aerosol. To get an indication how well the model reproduces the vertical properties of different aerosol compounds, we repeat the model evaluation of Koch et al. (2009) where Aerocom models were compared against observed BC concentrations from several aircraft measurement campaigns shown in Figure 12. Data from the following campaings were used: ARCPAC (Brock et al., 2011), ARCTAS (Jacob et al., 2010), ARCTAS-CARB (Jacob et al., 2010), TC4 (Toon et al., 2010), CR-AVE (https://espo.nasa.gov/ave-costarica2/), and AVE-Houston (https://espo.nasa.gov/ave-houston).

In addition, we evaluated the modeled mass concentrations of SU and OA measured in 17 different aircraft campaigns which have been compiled by Heald et al. (2011) shown in Figures 13 and 14. We also repeated the evaluation for the M7 and M7default setups. Data from the following campaigns were used: ACE-Asia (Huebert et al., 2003; Maria et al., 2003; Gilardoni et al., 2007), ADIENT (Morgan et al., 2010), ADRIEX (Highwood et al., 2007; Crosier et al., 2007), AMMA (Redelsperger et al., 2006; Capes et al., 2009), ARCTAS (Jacob et al., 2010; Cubison et al., 2011), DABEX (Haywood et al., 2008; Capes

et al., 2008), DODO (Capes et al., 2008), EUCAARI (Kulmala et al., 2009; Morgan et al., 2010), IMPEX (Dunlea et al., 2009), ITCT-2K4 (Heald et al., 2006; Sullivan et al., 2006), ITOP (Fehsenfeld et al., 2006; Lewis et al., 2007), OP3 (Hewitt et al., 2010; Robinson et al., 2011), TexAQS (Parrish et al., 2009; Bahreini et al., 2009), TROMPEX (Heald et al., 2011), VOCALS-UK (Wood et al., 2011; Allen et al., 2011).

Figure 12 shows the vertical profiles of BC concentration (black curve) measured using the Single Particle Soot Photometer (SP2, Droplet Measurement Technologies, Inc., Boulder, CO) on board of aircrafts. In this comparison, we used only the model data for the year 2010.

The red curves represent the monthly mean BC concentrations sampled along the flight path from the SALSA2.0 simulations. The monthly means were calculated for the year 2010 for the month during which each aircraft campaign was performed. The BC aircraft campaigns can be divided between campaigns in the tropics and midlatitudes (AVE Houston, CR-AVE, TC4, and CARB) and those performed at high latitudes (ARCTAS, ARCPAC). More details of these campaigns and their locations are given by Koch et al. (2009).

From Fig 12 we can see that near the source areas (tropics and midlatitudes) SALSA2.0 tends to overestimate BC concentrations quite significantly with the exception of the CARB campaign, where SALSA2.0 simulated BC concentrations are slightly lower than the observed mean and fall within the standard deviation of the data. Overestimation near the source areas can partly be attributed to the multiplication of biomass burning emissions by the factor of 3.4. In contrast, over high latitudes, SALSA2.0 simulated BC concentrations always fall below the observed mean. This is in line with many of the Aerocom models analyzed in the study by Koch et al. (2009).

Modelled SU and OA profiles showed significantly better comparison with the observations than BC. Especially the vertical profiles of SU in ACE-Asia, ADRIEX, TexAQS, EUCAARI, ARCTAS Summer, ITOP, and VOCALS-UK campaigns are captured very well by the model. The SU profiles for the campaigns are shown in Figure 13 and OA profiles in Fig 14. The coloured lines represent the average of model daily means sampled along the flight tracks and the corresponding days of the flights. For BC, the difference between the observations and the model was more than one order of magnitude, whereas for SU and OA the difference is in most cases significantly smaller. In many cases, modelled BC concentrations exceeded the limits of the variability of observations (grey whiskers in Figs 12, 13, and 14). However, modelled SU and OA concentrations fall within the variability of the observations in most campaigns. Note also that in Figure 12 concentrations are shown on a logarithmic scale, while in Figures 13 and 14 the scale is linear.

From these figures we can see that also for M7 the comparison between the model and the observations is clearly better for SU and OA than for BC. Similar to SALSA2.0, M7 tends to overestimate BC concentrations near the source regions while underestimating them at high latitudes. It is noteworthy that the simulated BC mass in SALSA2.0 and M7 generally agrees better near the surface and near the source regions than aloft and in the remote regions. At higher altitudes, above the $200\,\mathrm{hPa}$ pressure level, SALSA2.0 has always higher BC mass compared to M7. For the ARCPAC and Spring ARCTAS campaigns SALSA2.0 also simulates higher BC mass through the vertical column than M7. These differences indicate that in SALSA2.0, microphysical aging of BC is slower, which means that it takes a longer time for BC particles to obtain enough condensed material to be transferred to the soluble size classes in which they would be more efficiently removed.

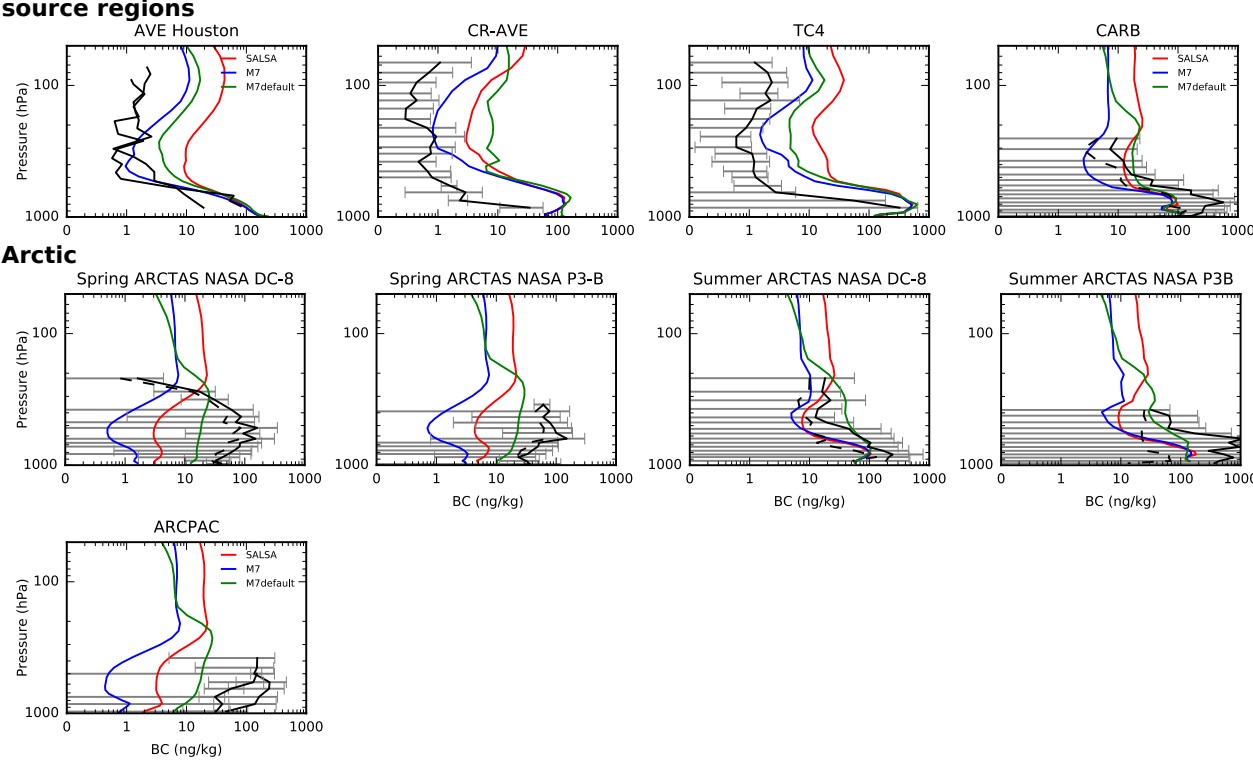

**Figure 12.** Observed and modelled mean vertical profiles of BC in aircraft measurement campaigns. Black curves represent the measured BC concentrations and the grey whiskers show the variability of measurements.

Since SU and OA masses are less sensitive to microphysical processing than BC, similar systematic differences are not seen between SALSA2.0 and M7 simulated profiles of SU and OA. On the contrary, SALSA2.0 and M7 profiles are very similar for most of the campaigns and in most regions SALSA and M7 differ much less with each other than both with the observations. Although the microphysical processing of SU was shown to produce different mass size distributions of SU between SALSA and M7 in Figure 4, this does not translate to differences in mass as it is not very sensitive to aerosol microphysics.

The new wet deposition scheme improves noticeably the comparison between the model and the observations from the Arctic campaigns. Comparing M7 to M7default, the differences are larger for the BC profiles than for SU and OA profiles, which are very similar for all three model setups. Especially for the ARCPAC and Spring ARCTAS campaigns the difference in BC concentration profiles between the two M7 setups becomes extremely large, with the difference being approximately two orders of magnitude near the ground level. This comparison is a clear indication that in order to simulate the vertical profiles of BC realistically, especially in the remote regions, an accurate description of both microphysics and wet deposition is required. This was also shown by Bourgeois and Bey (2011) who evaluated the effect of scavenging rates on the simulated Arctic BC concentrations and by Kipling et al. (2016) who compared the contribution of different aerosol processes on vertical profiles.

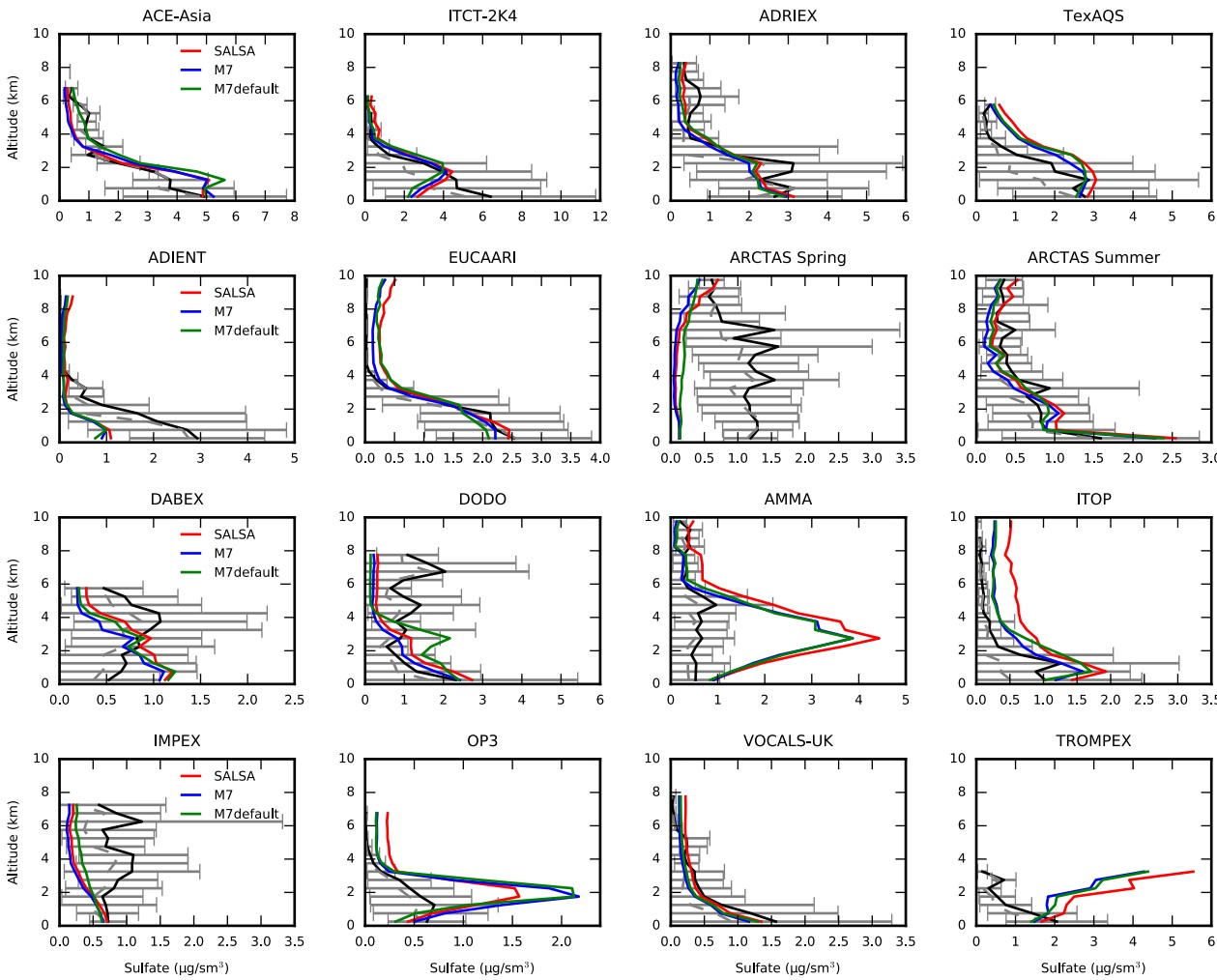

**Figure 13.** Observed and modelled mean vertical profiles of SU in aircraft measurement campaigns. Black curves represent the measured SU concentrations and the grey whiskers show the variability of measurements.

It has to also be noted that the model data was for different years than the observations. To see how much this affects the results, we did an additional comparison, where we used the exact years from the SALSA2.0 simulation. In this comparison, we used model values for the year 2010. For black carbon, we used the modelled monthly values from the flight path for the month of the year which correspondend to the observations. For sulfate and organic aerosol, we used the modelled daily values from the flight path for the day of the year which corresponded to the observations. The difference between using the whole simulation period of 2003-2012 and using the actual days of flights as opposed to using only one year was fairly small. For most campaigns and height levels, the relative difference in black carbon mass concentration is less than 50 % and the shape of the vertical profiles are very similar, mostly overlapping each other. The largest difference is for black carbon in the

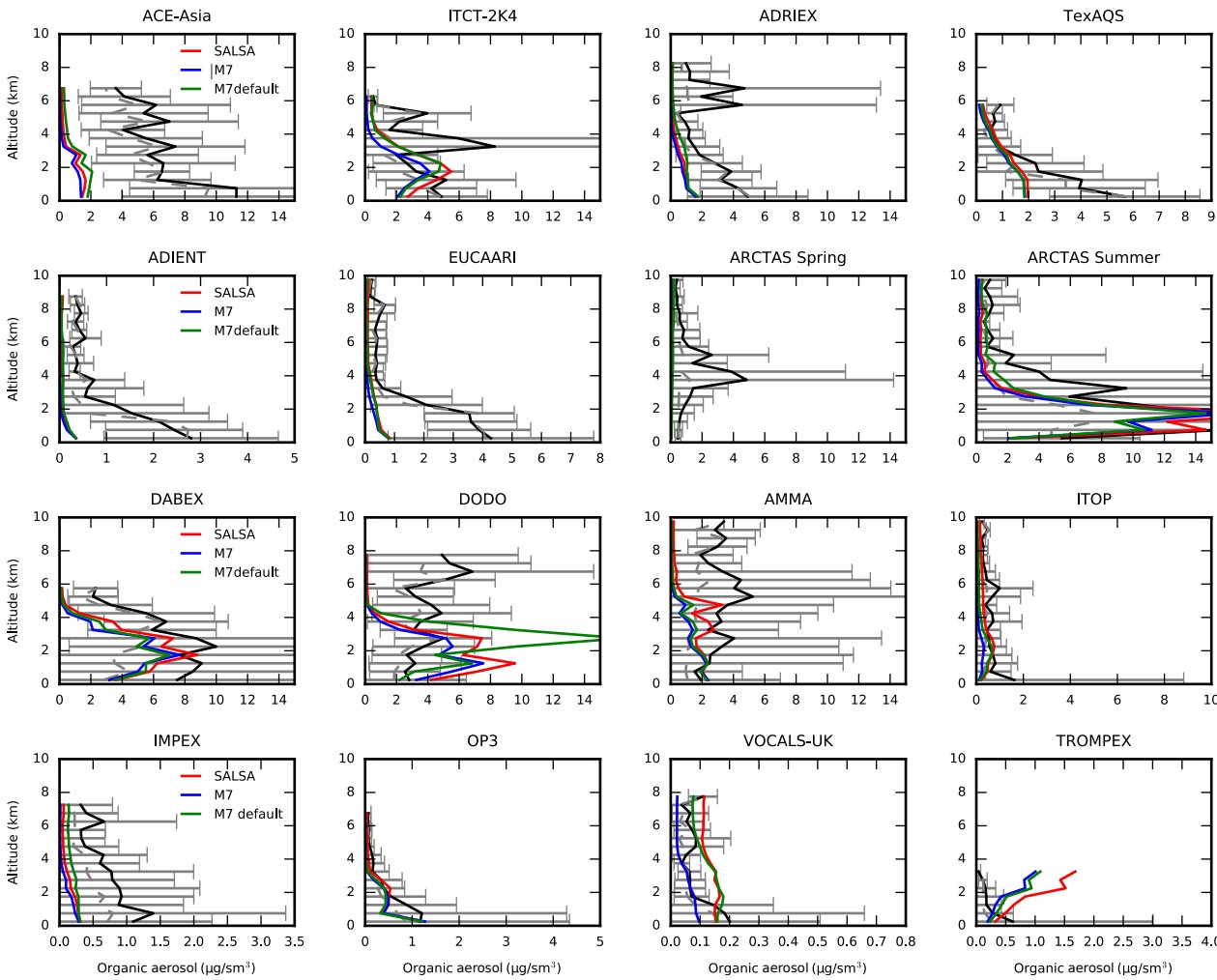

**Figure 14.** Observed and modelled mean vertical profiles of organic aerosol in aircraft measurement campaigns. Black curves represent the measured OA concentrations and the grey whiskers show the variability of measurements.

CR-AVE campaign where the relative difference in the mass concentration is ∼83 % in the lowest layer. However, the main characteristics of vertical profiles for all compounds were similar and would not change our conclusions.

## 5   Aerosol size distributions

Since the choice of the aerosol microphysical module will affect the particle properties that are most sensitive to microphysical processing, i.e. the number and the composition of fine particles, we evaluated the simulated size distributions. This was done by comparing the size distributions from SALSA2.0 and M7 simulations against those measured at the EUSAAR sites (Asmi et al., 2011). Figure 15 shows the median number and mass size distributions for four selected sites: Hyytiälä (boreal

region), Mace Head (marine), Zeppelin (Arctic), and Kosetice (industrialized). The figure is separated into four panels, each of which includes four subplots. In each panel, the upper row shows the yearly median number size distribution together with the EUSAAR observed number size distribution shown (blue solid curve) and the lower row shows the mass size distributions (bar plots). In each panel, the left column is for SALSA2.0 and the right is for M7. In order to make the comparison clearer for the reader, we remapped the M7 modes to SALSA2.0 size classes. All size classes also show the relative mass contribution of individual model compounds using colored bars. The EUSAAR measurements were made using either Differential Mobility Particle Sizers (DMPS) or Scanning Mobility Particle Sizer (SMPS) for which the measured size range corresponds roughly to the size range of SALSA2.0 ($\sim 3\,\mathrm{nm}$–$10\,\mu\mathrm{m}$). For the details about the measurements, see Asmi et al. (2011).

From Figure 15 we can see that both models reproduce the observed size distributions fairly well except for the Zeppelin station. The observed size distribution at Zeppelin exhibits a distinct mode with mean diameter of $\sim 0.2\,\mu\mathrm{m}$. This mode is not seen in either of the model setups. An overall difference between SALSA2.0 and M7 can be seen in the accumulation mode which peaks (both mass and number) at smaller sizes in SALSA2.0. Similarly to what was shown earlier in the MODIS comparison, this is likely because in the modal method, condensing species accumulate more the accumulation mode than in the sectional method (Zhang et al., 1999). In all four cases, the sulfate mass peaks in M7 in particles with diameters between 0.2 and $0.4\,\mu\mathrm{m}$. In SALSA2.0 the peak of sulfate has more station to station variation, but it also peaks in particles with diameters between 0.2 and $0.7\,\mu\mathrm{m}$. In general, the differences between the two model approaches are largest for sizes smaller that $0.2\,\mu\mathrm{m}$, i.e. the sizes that are more sensitive to microphysical processing.

## 6  Evaluation of the Pinatubo simulation

The simulations of the stratospheric aerosol formation and growth following the Mt Pinatubo eruption were compared against the High-resolution Infrared Radiation Sounder (HIRS) (Baran and Foot, 1994) and Raman lidar observations (Ansmann et al., 1997). Figure 16a shows the time evolution of the global burden of SU and $SO_2$ retrieved from HIRS, SALSA2.0, and the two M7 setups: one using the standard mode widths (M7), one using the mode widths recommended by Kokkola et al. (2009) (M7strat), and additionally one simulated using MAECHAM5-SALSA (denoted as SALSA1). The colored solid lines show the mean of the model ensemble and the shading the variability of different ensemble members. It has to be noted that the standard mode setup of M7 was optimized for describing tropospheric aerosol and was not intended to be used in the stratosphere. However, it is possible to modify the mode properties so that the model can simulate both tropospheric and stratospheric aerosol and has been done successfully in e.g. the GLOMAP-MODE global aerosol model (Dhomse et al., 2014).

From the figure we can see that here the difference between the sectional and modal setups becomes large. The simulated sulfur burden in the SALSA2.0 simulation is more than two times higher than in the M7 simulation at approximately 10 months from the start of the eruption. Provided that the estimate of the emissions strength of $8.5\,\mathrm{Tg}$ of sulfur for the Mt Pinatubo eruption is realistic SALSA2.0, SALSA1, and M7strat overestimate the sulfur burden in the early part of the simulation On the other hand, M7 in its standard mode setup underestimates the sulfur burden for most of the duration of the simulation period. This is because the effective radius of the particles in the standard M7 grows to larger sizes than in SALSA and the

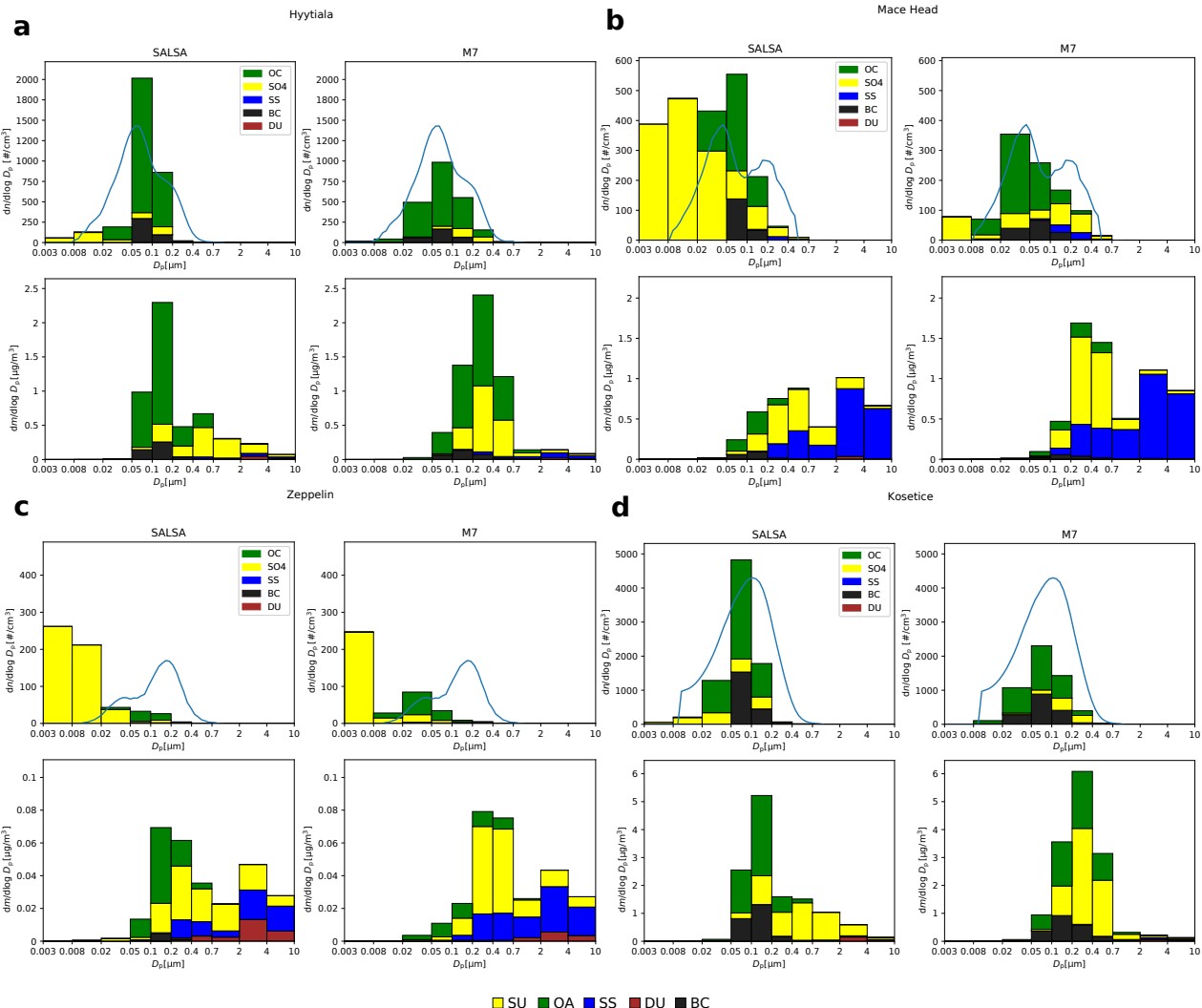

**Figure 15.** Observed and modelled yearly median number size distributions ($\mathrm{d}n/\mathrm{d}\log D_\mathrm{p}$), and modelled mass ($\mathrm{d}m/\mathrm{d}\log D_\mathrm{p}$) size distributions for four different EUSAAR stations: a) Hyytiälä, b) Mace Head, c) Zeppelin, and d) Kosetice. Observed values are represented by the solid blue curves and the observations are represented by the bar plots. The relative mass contribution of individual chemical compounds in each size class are denoted by a color.

growth enhances the removal of stratospheric sulfate particles by sedimentation. In the simulated volcano plume, growing sulfate particles form a mode of a very narrow width with a diameter of approx $1\,\mu\mathrm{m}$ (Kokkola et al., 2008) which is not well represented by the standard M7 coarse mode which has the geometric mean deviation of 2. The stratospheric aerosol configuration of M7 (M7strat) brings the M7 values close to SALSA2.0 values, however this mode setup is then only valid for

5    the volcanic plume and very likely decreases the models ability to simulate the tropospheric aerosol. Especially the simulated tropospheric coarse mode particles are very different between the standard M7 and M7mod. For example, the annual global

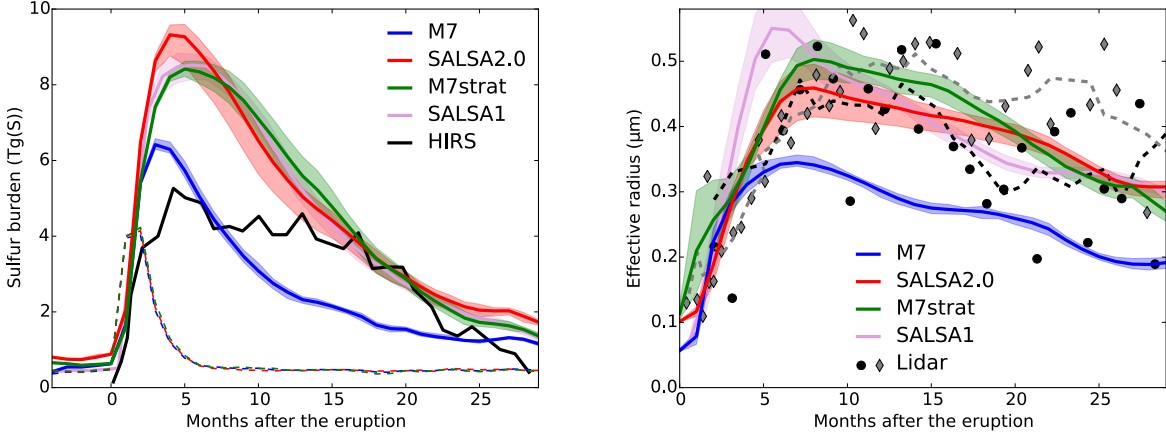

**Figure 16.** a) The simulated global burden of $SO_2$ (dashed curves) and sulfate (solid curves), simulated by SALSA2.0 (red curves), M7 (blue curves), M7strat (green curves), and SALSA1 (purple curves). The black curve shows the global sulfate burden retrieved from HIRS observations. The shading around the solid curves represents the variability of the model ensembles. b) The mean effective radius of stratospheric aerosol in the 12 km to 20 km layer observed by Raman lidar (diamond markers represent the observations at Laramie, circles represents those at Geesthacht, dashed lines show the 5 month running means), simulated by SALSA2.0 (red curves), M7 (blue curves), M7strat (green curves) and SALSA1 (purple curves). The shading around the solid curves represents the variability of the model ensembles. .

burden of mineral dust is 6 Tg in the M7mod simulation, which is less than 40 % of the values for the standard M7 simulation values shown in Table 3 being near the lower limit of the estimate by (Liu et al., 2005) and are likely to be significantly underestimated.

The evolution of the effective radii of the stratospheric aerosol after the eruption was also evaluated against Raman lidar re-
trievals from balloon-borne observations at Laramie, Wyoming (41° N, 141° W) and ground based observations at Geesthacht, Germany (53° N, 10° E). Figure 16b shows the mean effective radius in the 12 km to 20 km layer observed by the Raman lidars as well as those simulated using SALSA2.0, M7 in its standard mode configuration, and M7 with the modified mode setup.

From Fig 16b we can see that SALSA2.0 reproduces the retrieved values of the effective radii, which do have a very large
variability. The effective radii in the SALSA2.0 simulations follow the mean of the retrieved values as well as the time evolution of the retrieved effective radii. In M7, the coarse mode is effectively removed from the stratosphere and thus the effective radius is underestimated and the M7 values are at the low end of the retrieved values. M7mod setup shows a much better comparison following the average of the lidar measurements. The effective radius in M7mod reaches a larger maximum value and decreases faster than in SALSA2.0. Out of the two, SALSA2.0 corresponds slightly better to the observed time evolution of the effective
radius. SALSA1 simulation results are fairly similar to both SALSA2.0 and M7strat results. Especially, the sulfur burden of SALSA1 follows closely the values in the M7strat simulation. The effective radius in SALSA1 peaks earlier and higher than in SALSA2.0 and M7strat simulations.

## 7    Conclusions

We coupled the sectional aerosol module SALSA2.0 to the aerosol-chemistry-climate model ECHAM-HAMMOZ. During the coupling, also HAM (the aerosol model of HAMMOZ) was modified to implement the sectional aerosol model SALSA2.0 as alternative to the default modal microphysics module M7. ECHAM-HAM coupled with SALSA2.0 was evaluated using a 10 year simulation period for the years 2003-2012, preceded by a one year spin up. Using 3-hourly output, SALSA2.0 required double the calculation time of M7 with Cray XC-30 architecture when 120 cores were used for the simulation. Simulated aerosol optical depths were evaluated against those retrieved from satellite-based MODIS instruments and ground-based AERONET sun photometers. Aerosol mass concentrations of individual compounds were evaluated against EMEP and IMPROVE networks of ground-based particulate mass concentration observations and vertical profiles from several different aircraft campaigns.

The aerosol optical depths simulated with ECHAM-HAMMOZ-SALSA2.0 were biased slightly low compared to both MODIS and AERONET retrievals. Local differences were the highest over Southern Hemisphere oceans, deserts, southeast Asia, and regions affected strongly by biomass burning . Over the oceans and deserts, these differences are very likely caused by emissions of natural aerosols. In desert regions, dust emissions are very sensitive to the model meteorology as they are driven by the simulated 10 meter wind speed (Bergman et al., 2012). Currently, the dust source strenghts in ECHAM-HAMMOZ, are optimized for M7 and thus a better match between the observations and SALSA2.0 could be achieved by optimizing the source strenghts for SALSA2.0. Over the Southern Hemisphere oceans, the newly introduced sea salt emission scheme (Long et al., 2011) is likely the main cause of the underestimation in AOD (in both SALSA2.0 and M7) as the simulated SS mass concentrations are much lower than with the emission parameterization of Guelle et al. (2001) which was used in the previous version of ECHAM5-HAM. The overestimation of AOD over biomass burning regions indicates that in this model configuration using the multiplier 3.4 for GFASv1 emissions produces excessive aerosol load near the sources. Over southeastern Asia, the reason for the low bias in SALSA2.0 simulated AOD against observations is likely due to the aerosol microphysical processing of the aerosol size distribution. This conclusion is backed up by the fact that M7 overestimates AOD over the same reason despite having the same emissions over that region as SALSA2.0.

When comparing the AODs simulated by SALSA2.0 and M7, the largest differences between the model versions occur in regions where SALSA2.0 also differs most significantly from the observation, i.e. deserts, southeast Asia and regions affected by biomass burning. The differences in southeast Asia and biomass burning regions are mainly caused by the different microphysics schemes as in these region, the size distribution is heavily modified by the condensation of sulfuric acid on aerosol. The methods for solving gas-to-particle partitioning in M7 and SALSA2.0 are different. In M7, the method presented by Kokkola et al. (2009) is used while SALSA2.0 uses the method presented by Jacobson (2005). In addition, the choice of chemical compounds that are taken into account in different size classes and modes cause differences between SALSA2.0 and M7. In M7, the insoluble accumulation and coarse modes do not include organic compounds while in SALSA2.0, organic compounds are included in all size classes. This results in different the composition size distributions of organics in the two model configurations.

The evaluation of aerosol mass concentrations against surface measurements showed that simulated SU mass concentration on average exceeds the observed SU mass while OA and BC are slightly underestimated. This holds for both EMEP and IMPROVE stations. Simulated DU and SS were underestimated in the majority of stations. Especially, the simulated SS was significantly underestimated indicating that the Long et al. (2011) emission parameterization used in the SALSA2.0 configuration is biased low, a conclusion also supported by low AOD over the oceans. In the comparison between the SALSA2.0 simulated vertical profiles of SU and OA mass concentrations were in fairly good agreement with those measured in aircraft campaigns. However, the vertical profiles of BC mass concentrations in the SALSA2.0 simulations and aircraft measurements had large discrepancies especially in the Arctic with differences of more than one order of magnitude.

Comparison to M7 simulations showed that the vertical profiles of SU and OA are not very sensitive to the choice of the microphysics module. However, the simulated vertical profiles of BC mass concentrations show fairly large difference between SALSA2.0 and M7 especially at high altitudes and away from the source regions. This is likely to be caused by differences in the rate of microphysical aging of BC. However, in the current ECHAM-HAMMOZ version, SALSA2.0 and M7 simulated SU and OA vertical profiles seem to be more similar than in the previous model version (Kipling et al., 2016).

Simulated size distributions show fairly good comparison against those measured at EUSAAR sites. Especially compared to the previous generation model version ECHAM5-HAM-SALSA1 (Bergman et al., 2012), the agreement between measured and modelled size distributions has improved. ECHAM5-HAM-SALSA1 showed significant underestimation of particle numbers at most EUSAAR stations (Bergman et al., 2012), in the current model version such bias is no longer evident.

Overall, the microphysical scheme affects mainly particles in the lower end of the size spectrum, the simulated number size distributions and mass size distributions in SALSA2.0 and M7 differ especially for sizes smaller than 0.7 µm. The largest difference among different model compounds is in the accumulation size mass distribution of SU which is the only compound affected by condensation. One reason for this discrepancy is that modal models tend to overestimate the condensational growth of accumulation size particles (Zhang et al., 1999).

One simulated case where SALSA2.0 reproduces the observations considerably better than the default tropospheric setup of M7 is the simulation of the volcanic plume produced by the 1991 Mt Pinatubo eruption. In the volcano plume, microphysical processes affect strongly the aerosol size distribution leading steep gradients at particle sizes of approximately 1 µm in diameter. This is because the sectional size distribution allows for steep gradients in the size distribution. Such steep gradients are evident in volcanic plumes as the condensation of sulfuric acid "narrows" the size distribution by growing small particles faster than the largest ones.

Overall, SALSA2.0 performs slightly better than M7 in the evaluation cases where the statistical metrics were possible to calculate. Out of 9 comparisons against the observations of optical properties and mass, in 6 of them SALSA2.0 had smaller root mean square deviation, in 5 it had a smaller normalized mean bias and in 7 it had a higher value in the correlation coefficient R. On the other hand, it has to be noted that for many aerosol properties, e.g. vertical profiles of SU and OA mass concentration, SALSA2.0 and M7 show better agreement between each other than with the observations.

The result of this study indicate that SALSA2.0 is a competitive choice for modal aerosol microphysics modules in global atmospheric models. A sectional scheme, such as SALSA2.0, can capture a wide variety of possible atmospheric size distri-

butions, including explosive volcanic eruptions and stratospheric geoengineering for which modal models often need tuning. Size-dependent anthropogenic aerosol emissions, which are starting to become available (e.g., Xausa et al., 2017), can also be easily incorporated into sectional modules further improving the ability of SALSA2.0 to realistically reproduce global aerosol size and composition.

5   *Code availability.* The ECHAM6-HAMMOZ model is made available to the scientific community under the HAMMOZ Software Licence Agreement, which defines the conditions under which the model can be used. The licence can be downloaded from https://redmine.hammoz. ethz.ch/attachments/download/291/License_ECHAM-HAMMOZ_June2012.pdf. The standalone zero dimensional version of SALSA2.0 is distributed under the Apache-2.0 licence and the code is available at https://github.com/UCLALES-SALSA/SALSA-standalone/releases/ tag/2.0 with DOI 10.5281/zenodo.1251668

10   *Data availability.* The model data can be reproduced using the model revision r4098 from the repository https://redmine.hammoz.ethz.ch/ projects/hammoz/repository/show/echam6-hammoz/branches/fmi/fmi_trunk. The settings for the simulation are given in the same repository, in Folder gmd-2018-47. MODIS data is available for download from Level 1 and Atmosphere Archive and Distribution System (LAADS) https://ladsweb.modaps.eosdis.nasa.gov/search/. AERONET data can be obtained using the Aerosol Robotic Network download tool https:// aeronet.gsfc.nasa.gov/cgi-bin/webtool_opera_v2_new. EMEP data is available for download from the EBAS database at http://ebas.nilu.no/. 15   IMPROVE data is available for download from the Federal Land Manager Environmental Database http://views.cira.colostate.edu/fed/ DataWizard/Default.aspx. AEROCE and SEAREX data can be downloaded from http://aerocom.met.no/download/DUST_BENCHMARK_ HUNEEUS2011/conc_aeroce.prn. The aircraft measurement data for BC can be downloaded from http://aerocom.met.no/download/BC_ BENCHMARK_KOCH2009/. EUSAAR size distributions are available for download at https://www.atm.helsinki.fi/eusaar/. The data for the Mt. Pinatubo evaluation can be downloaded from https://redmine.hammoz.ethz.ch/projects/hammoz/repository/show/echam6-hammoz/ 20   branches/fmi/fmi_trunk/gmd-2018-47 . The aircraft data for SU on OC was received from several measurement teams who hold the ownership for the data and thus it is only provided by request from harri.kokkola@fmi.fi.

*Competing interests.* The authors declare that no competing interests are present.

*Acknowledgements.* The ECHAM-HAMMOZ model is developed by a consortium composed of ETH Zürich, Max Planck Institut für Meteorologie, Forschungszentrum Jülich, University of Oxford, the Finnish Meteorological Institute and the Leibniz Institute for Tropospheric 25   Research, and managed by the Center for Climate Systems Modeling (C2SM) at ETH Zurich. This work was supported by the Nordic Center of Excellence, eSTICC (57001), the Academy of Finland Center of Excellence programme (grant no. 307331), the Academy of Finland research projects no. 308292, 283031, and 287440, the European Research Council (ERC) Consolidator Grant No 646857, and the European Union's Horizon 2020 research and innovation programme under grant agreement No 641816 Coordinated Research in Earth Systems and Climate: Experiments, kNowledge, Dissemination and Outreach (CRESCENDO). Philip Stier would like to acknowledge funding from the

European Union's Seventh Framework Programme (FP7/2007-2013) projects BACCHUS under grant agreement 603445 and the European Research Council project ACCLAIM under grant agreement FP7-280025 as well as the European Research Council project RECAP under the European Union's Horizon 2020 research and innovation programme with grant agreement 724602. We thank Hauke Schmidt and Sebastian Rast (Max Planck Institut für Meteorologie) for their work on developing the ECHAM model and the interface between ECHAM and HAMMOZ, Maria Kanakidou (ECPL, University of Crete) for the help in compiling the EMEP and IMPROVE datasets. For the aircraft data we thank Hugh Coe (University of Manchester), Lynn Russell (Scripps Institution of Oceanography), Rodney Weber (Georgia Institute of Technology), Jose Jimenez (University of Colorado at Boulder), Roya Bahreini (University of Colorado - CIRES, NOAA ESRL Chemical Sciences Division), Ann Middlebrook (NOAA ESRL Chemical Sciences Division), James S. McDonnell Foundation Award for 21st Century Science, NOAA grant NA17RJ1231, National Science Foundation grants ATM-0002035, ATM-0002698, and ATM04-01611, and the NERC Global Aerosol Synthesis and Science Project (GASSP) NE/J023515/1. IMPROVE is a collaborative association of state, tribal, and federal agencies, and international partners. US Environmental Protection Agency is the primary funding source, with contracting and research support from the National Park Service. The Air Quality Group at the University of California, Davis is the central analytical laboratory, with ion analysis provided by Research Triangle Institute, and carbon analysis provided by Desert Research Institute.

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
