# Peer review of "SALSA2.0: The sectional aerosol module of the aerosol-chemistry-climate model ECHAM6.3.0-HAM2.3-MOZ1.0"

_Geoscientific Model Development, 2018_

## Short Comment (SC1) · 27 Apr 2018

Thanks for your manuscript. When code availability is referencing Github the model version should be marked as a release. There seems to be a release 1.0 available only. Also authors are encouraged to create a DOI for the release and cite this in the paper. For code in Github a DOI can easily be created through Zenodo, see https://guides.github.com/activities/citable-code/ for details. Lutz Gross GMD Executive Editor

---

## Referee Comment (RC1) · Anonymous Referee #1 · 10 May 2018

In this study an updated version of the SALSA sectional aerosol module in the ECHAM-HAMMOZ is described. In an extensive comparison the model results are compared to MODIS satellite retrievals, AERONET measurements, measurements from the IM-PROVE and EMEP network, multiple aircraft campaigns and size distribution measurements at EUSAAR sites. Performance is compared to the HAM default aerosol scheme M7. Simulations of the Pinatubo are added to demonstrate the performance of SALSA2.0 in atypical aerosol regimes. The validation against measurements is very comprehensive and addresses many different aspects of the aerosol distribution. However, by limiting the comparison of the performance of SALSA2 to M7 it is difficult to judge if SALSA2.0 is a real improvement to SALSA1 or to judge how SALSA2.0

performs compared with other sectional aerosol modules which have a fundamentally different approach and substantially increase computational costs. Although the following points might involve a lot of work, addressing these would greatly improve the quality of this work.

**1   General comments**

1. Performance of SALSA2.0 is compared to the ECHAM-HAMMOZ default (modal) aerosol module M7. Consequently, many of the differences in this work are attributed to the difference in the numerical treatment of the aerosol size distribution in modal or sectional schemes. Because of this, the manuscript does not really give an indication of how well SALSA2.0 performs compared to other sectional modules.

2. The paper describes the difference between SALSA1 and SALSA2.0 well, but does not discuss the reasons why certain changes are made to the SALSA module. Please explain what the main problems with SALSA1 are and how these changes contribute to the improvement of the aerosol module.

3. Previous work (also referred to in this article) has shown that modal aerosol representations do not perform well in simulating stratospheric aerosol caused by volcanic eruptions and that sectional approaches yield far better results. It is therefor not surprising that SALSA2.0 performs better than M7, but how does it compare to e.g. SALSA1? Overall, the discussion of the Pinatubo simulation is thin and mainly addresses issues in M7. As it is now, it might be better to remove this section from the paper as the remainder is already a very comprehensive comparison to observations.

4. The authors state that a size-resolved wet deposition scheme for SALSA2.0 is still

under development. To make a fair comparison to M7 an older removal scheme is also used for those simulations. In my opinion, the quality of this paper would greatly improve if the new wet deposition scheme is included in this work. In the comparison to aircraft measurements in the results section it is also explicitly mentioned how the new wet deposition scheme would improve the results of the simulations with SALSA2.0.

**2 Specific comments**

**Title / Page 8, line 9-13** Judging from the text, the MOZ module is not used in this work and HAMMOZ is reduced to HAM only. This is a bit misleading and causes confusion in the text. It is not clear what value the combination of HAM and MOZ has in this work. It should be clarified better what MOZ does in the model simulations for this work, otherwise it might be better to remove MOZ from the title.

**Page 2, line 31** 100 size classes is a bit of an exaggeration, in global models the number of size classes is usually (much) lower.

**Page 5, line 21-23** Two subranges of SALSA1 are combined into one in SALSA2.0, what is the reason for this simplification and what are expected changes in the simulated aerosol size distribution?

**Page 5, line 25-26** The moving center method is replaced by the hybrid bin method. What are the downsides to this method as is was not used in SALSA1 before?

**Page 7, line 8-9** In view of the importance of meteorology for e.g. dust emissions, what is the nudging time interval used in the simulations?

**[GMDD](GMDD)**
**Page 8, line 14-34** The different parameterisations of SALSA2.0 and M7 are explained extensively in Section 2.3, but several of these are changed for the simulations. This is very confusing and makes large parts of previous section irrelevant. It would be better to describe what is actually used. Also, this means that this work presents results from a suboptimal model run and the full potential of the SALSA2.0 module in the ECHAM-HAMMOZ model is not shown.

**Page 9, line 26** How is this ensemble constructed? What is the difference between the 5 members?

**Page 13, line 6** Judging from this work, the implementation of the new Long et al. (2012) sea salt emission decreases the model performance, why was it introduced?

**Page 15, line 4** Here, the low AOD bias is (almost) completely attributed to the low SS emissions. Although this assumption is acceptable for the SH, there is also a strong bias in the NH high latitudes. Here, the low bias over the land masses cannot be attributed to sea salt only.

**Page 15, line 13-14** How did you arrive to this conclusion?

**Page 15, line 25** Why is this not mentioned in Section 4.1.1?

**Section 4.2** Restructure section. Multiple species of multiple model runs are compared to multiple measurement networks, This already makes the discussion hard to read. I suggest a fixed format/structure in discussing the different species to help the reader.

**Page 18, line 1-22** Include comparison results of M7 in discussion of SU/BC.

**Page 19, line 9** Aerosol load over oceans is not low over SH subtropics.

**Page 19, line 15** If periods in observations and simulations are different, how are they collocated?

**Page 22, line 20** Why are monthly mean model values used here? Having 3 hourly output, collocation can be greatly improved. Also, comparison for SU and OA is based on daily mean output. What is the reason for this inconsistent approach?

**Page 25, line 5-6** This conclusion is too strong and drawn too quickly. It would be the case for the ARCTAS Spring and ARCPAC campaigns, but for the ARCTAS Summer, the wet deposition scheme barely influences the results for the lower part of the atmosphere where observations are available. Also, for the source regions, an increase due to the wet deposition scheme, would increase the already high bias in SALSA.

**Page 26, line 1** "Difference was fairly small". Can this statement be quantified?

**Section 6** In this section, it is explained why the section approach of M7 does not perform well in simulation the stratospheric aerosol burden resulting from a volcanic eruption. There is even a reference to a solution for this problem. Yet you don't incorporate this in your model and compare the performance of SALSA2.0 mainly to the simulation with the unadjusted M7 scheme. As a result, it is difficult to really judge how well SALSA2.0 performs in these simulations. It would be more interesting to see the performance of SALSA2.0 to other sectional aerosol modules or modal schemes that were properly adjusted.

**Page 26, line 1-2** How are the model values and observations collocated?

**Section 7. Conclusions** The structure of the conclusion section is unnecessarily confusing. Follow same order as discussion in Results section.

**Page 29, line 6** What are recommendations for optimizing?

**Page 30, line 1-2** Underestimation of particle number in SALSA1 not mentioned in Section 4.1.1.

**3   Technical comments**

**Page 4, line 12**  Sentence is not clear, please rephrase.

**Page 5, line 13**  "using the volume ratio" → "using volume ratio"

**Page 7, line 10**  Add full name of PCMDI

**Page 10, line 21-23**  Why is India omitted from this list?

**Page 11, Figure 2**  Add statistics (e.g. corresponding global mean AOD to a,b,c and correlation coefficients and NMBs to d,e) to the figure for a good overview between model configurations.

**Page 12, line 2**  Equation straightforward, can be removed and explained in words.

**Page 13, line 1**  Fig 4. → Fig. 5

**Page 14, line 12**  Reference to current section.

**Page 15, line 17**  Add minus sign to 0.05.

**Page 16, Figure 6**  Change colours of Asia and North America. These are the two regions discussed in the text but hardly distinguishable from each other. Also, adding a regional mean values would provide a good overview of model performance.

**Page 17, Figure 7**  Remove additional abbreviations of species in lower left corner of each panel. Add names of network to panels.

**Page 20, Figure 9**  Observed and simulated values in year 2010? Please add to caption.

**Page 22, line 32**  Vertical profiles of AMMA, ARCTAS Spring and OP3 are not captured well either.

**Page 26, line 1-2** Add references for the HIRS and lidar observations.

**Page 29, Fig. 14** Add errorbars to observed values.

**Page 30, line 23** confiburations → configurations

---

## Author Comment (AC1) · 23 May 2018

*Thanks for your manuscript. When code availability is referencing Github the model version should be marked as a release. There seems to be a release 1.0 available only. Also authors are encouraged to create a DOI for the release and cite this in the paper. For code in Github a DOI can easily be created through Zenodo, see https:// guides. github.com/ activities/ citable-code/ for details. Lutz Gross GMD Executive Editor*

Thank you for these suggestions. We have now modified the GitHub release numbering to reflect the versioning in the manuscript title. Release 2.0 can be found here: https: //github.com/UCLALES-SALSA/SALSA-standalone/releases/tag/2.0

[Figure]

We have also obtained a DOI for the release which is: 10.5281/zenodo.1251668

We will provide the DOI and the link to the GitHub release version in the revised manuscript

---

## Referee Comment (RC2) · Anonymous Referee #2 · 9 Jun 2018

This manuscript describes the implementation of a sectional aerosol microphysics module (SALSA2.0) within the composition-climate model ECHAM-HAMMOZ (ECHAM6.3.0-HAM2.3-MOZ1.0), as an alternative to the existing modal aerosol microphysics scheme "M7". The paper then evaluates aerosol optical properties, aerosol mass and particle size distribution simulated by ECHAM-HAMMOZ-SALSA, comparing to observations, and to aerosol properties simulated with the composition-climate model in its usual configuration with the existing aerosol scheme (ECHAM-HAMMOZ-M7).

The topic is certainly within the scope of Geoscientific Model Development, and the description of the implementation into the model will be a valuable resource for those applying the model. The paper is organised well, and the comprehensive set of comparisons to different observed aerosol properties provides a good test of the new ECHAM-HAMMOZ-SALSA model.

However, some of the principal statements made in the Abstract are not supported by the results presented in the paper, and much more care needs to be taken in the statements interpreting potential reasons for differences between the sectional and modal schemes.

In particular, the authors claim that the size distribution comparisons shown in Figure 4, shown for locations where AOD difference is largest, are indicative that of different microphysical processing in the modal and sectional schemes, and that this is the reason why the two aerosol schemes predict different extinction/AOD in these regions.

But from careful inspection of Figure 4, it's clear that is not the case. The authors already identify the two locations (in China and in Russia) as regions where the observed AOD is very high, and clarify that the Russian site is in a region where biomass burning emissions are high. The China site is in a region of strong anthropogenic emissions.

The size distribution of the black carbon (the black bars in the stacked bar chart) are very different between the SALSA and M7 simulations at both locations and this clearly indicates that there is a systematic difference in the size at which primary carbonaceous aerosol particles are emitted, which is a much more likely explanation of the reason for the difference.

At the China site, the M7 run has about half of the BC in particles larger than 200nm, whereas for the SALSA run this is only about 10%. The same is true for the Russia site, indicating that there is a systematic difference between the two schemes in the sizes assumed for primary carbonaceous emissions.

This is an important issue, because if such large differences in AOD could indeed be

attributed to the simpler modal scheme having inadequate representation of microphysical processing compared to the bin scheme, then this could be cited in the literature extensively as a reason to justify the additional expense required for sectional aerosol schemes.

It is noticeable that whereas the Table 1 explains in detail the size segregation of the sea-salt and dust emissions, there is no information given about the assumed size at which the carbonaceous particles are emitted (yet these are the dominant primary aerosol in polluted regions).

I am sure this is just an oversight in the writing of the paper, and that there was no intention to omit this information, or to make a statement that is not supported by the results.

It is also clear from Figure 2 that, for many regions, the M7 scheme could actually be argued to perform better (compared to the MODIS AOD) than the SALSA scheme. SALSA seems to have substantial bias over North Africa and over marine regions in the Southern tropics, for example.

However, I am recommending major revisions to the paper, and request that the senior co-authors on the paper carefully go through the text with the first author, to check the interpreting statements being made. This should help ensure all statements made in the revised version are correctly interpreting potential reasons for the differences to the observations or the original modal scheme.

I have made a list of specific revisions for pages 1 to 13 which the authors need also to make to improve the first part of the paper.

Most of these are minor but the improved wording will help the reader to better understand the issues and specifics of the implementation.

One final general comment was that it needs to be stated somewhere early on in the text the difference between the acronyms "HAM" and "M7". My understanding is that

"HAM" is the overall modal aerosol scheme and that M7 is a component of HAM, basically the modal microphysical routines.

Please can the authors clarify I am understanding this correctly.

The reason I ask for this clarification is that I was expecting then the SALSA to not just be an alternative to M7, but an alternative to HAM, and that perhaps the correct naming should then be ECHAM6.3.0-SALSA2.0-MOZ1.0 when the SALSA scheme is applied.

However, perhaps that is not quite right and the implementation of SALSA into the model has in fact only implemented the microphysics routines within SALSA (or indeed that SALSA has always only been the microphysics routines).

I realise that within HAM there is a separate acronym for the microphysics routines (M7) than the overall modal framework, which is known as HAM.

By contrast many other aerosol schemes do not have this distinction and there is only one acronym for the overall aerosol module including both the microphysics routines and the other aspects (primary emissions, dry deposition, scavenging).

The naming convention of the different parts of the model are important in this case as it helps the reader to appreciate which aspects of the HAM scheme have been retained in the implementation of SALSA.

I realise different groups will have different ways of naming their modules – and I'm not necessarily suggesting the SALSA group come up with a new acronym for the microphysics elements of SALSA. However I do think it needs to be stated somewhere in the section 2.2 description exactly what constitutes the Hamburg Aerosol Model and what are the SALSA aspects. See also my first specific comment about the wording of the title.

Two principal major revisions required

———————————————

1) Abstract, page 1, lines 5-6 – As per my main comments above about Figure 4, this sentence is not supported by the results and needs to be removed or reworded. If the authors can repeat either the M7 or the SALSA simulations with the emissions size distribution for emitted carbonaceous particles identical in the two schemes then it may be possible to make some statement about this, but the different BC-size-distribution in M7 clearly indicates there is a substantial differece in the "emissions size distribution" applied for carbonaceous particles in the two aerosol schemes, which is much more likely reason for the difference in AOD between the two schemes. In any case the locations shown in Figure 4 (in Russia and China) are regions of very strong primary emissions. Lee et al. (2011) apply a perturbed parameter uncertainty analysis to show how (at least for the global aerosol microphysics scheme applied there) the regions where aerosol properties are most dominated by uncertainties in microphysical processes are away from such "emissions hot spots". So even if one of the models was re-run with the same "primary emissions size distribution" as for the other, one might expect any difference from microphysical processes to have most impact in a different region than the two locations shown.

2) Abstract, page 1, lines 9-11 – This difference in the modal and sectional aerosol microphysics predictions for the microphysical evolution and global dispersion of the Pinatubo volcanic cloud is interesting, but, as the authors point out, the standard mode widths for M7 are not intended to be applied to the stratospheric aerosol evolution. Figure 14 shows that actually, provided the model is applied with the "stratospheric-enabling adjustment" to the accumulation mode and coarse mode widths, the modal scheme compares well to the sectional scheme. As I understand it, the Hamburg stratospheric aerosol modelling group would not apply the model without this adjustment to the mode widths, so the emphasis really needs to be changed in how this is worded in the Abstract – and in the discussion of the results. I think it is very important, to minimise the chance of an incorrect inference from the reader, to present the

results having that "M7mod" essentially as the default (or even "validated"?) configuration for when the model is applied for simulating interactive stratospheric aerosol. Indeed I would strongly recommend to change the "branding" of that model run to "M7-strat" rather than "M7-mod". As I understand it, the adjustment to the mode widths is a pre-requisite for simulating stratospheric aerosol for that scheme, so the authors of the manuscript need to change the current wording of the results to be clearer that it is essentially "the stratospheric configuration of M7" or so. One could consider it in some ways equivalent to a tropospheric or stratospheric chemistry scheme. One would not apply a tropospheric chemistry scheme to simulate the chemistry of the stratosphere. I consider this another major revision required, and further explains why, although the paper is generally well-written and is a very good paper, I am recommending major revisions are required to more appropriately interpret differences between the predictions with the modal and sectional aerosol microphysics schemes.

List of minor revisions

————————

1) Title, page 1, The way the title is currently worded suggests the SALSA2.0 module is as a new sub-model within the Hamburg Aerosol Model (HAM). Is that the case then that HAM includes both M7 and SALSA as alternative aerosol microphysics modules? Or is the SALSA module an alternative to "the overall HAM" or so?

2) Abstract, page 1, 1st sentence – Related to point 1) is ECHAM-HAMMOZ still ECHAM-HAMMOZ when SALSA is applied – or should it then be referred to as ECHAM-SALSAMOZ or so?

3) Abstract, page 1, 2nd sentence – insert "aerosol" between "microphysics" and "alternative" to be clear it is aerosol microphysics not cloud microphysics.

4) Abstract, page 1, 3rd sentence – insert "within ECHAM" or "within ECHAM-HAMMOZ" between "implementation" and "is evaluated" to be clear it is this particular

implementation that is evaluated (one could imagine it potentially being implemented in another framework at some point in the future).

5) Abstract, page 1, 3rd sentence – delete "the" between "against" and "observations".

6) Abstract, page 1, 4th sentence – suggest this sentence be shortened and added to the end of the previous sentence (makes the Abstract easier to read). Specifically I'm suggesting replacing "distributions. We also compare the skill of SALSA2.0 in reproducing the observed quantities to the skill of the M7 implementation" with "distributions, comparing also to the skill of the M7 implementation".

7) Abstract, page 2, lines 1-2 – as per major comment 1-2, this sentence needs to be changed since (as I understand it) the M7 microphysics would not be applied for stratospheric aerosol applications unless the mode widths for the accumulation and coarse soluble modes were reduced to 1.2 in this way. In this sentence and the results of the Pinatubo comparisons, I this should be referred to as "the stratospheric aerosol configuration of M7" or similar.

8) Abstract, page 2, line 2 – I think a few additional qualifying words should be added re: why the mode widths need to be reduced 1.2. The size distribution measurements (Deshler et al., 2003) show that the shape of the accumulation mode for the particle size distribution in the mid-latitude stratospheric aerosol layer was consistent with a narrower standard deviation. I recommend to add the words ", as observed after Pinatubo (Deshler et al., 2003)." at the end of that sentence.

9) Introduction, page 2, line 6 – Suggest to add "global variation of" after "Describing the", delete "in these properties" at the end of the sentence, and insert "spatial and temporal" between "large" and "variability"

10) Introduction, page 2, line 7 – Suggest to replace "diameter of the particles can span" with "diameter of aerosol particles spans" (it's best to refer to "aerosol particles" rather than just "particles" to be clearest).

11) Introduction, page 2, lines 8-9 – The words "at the lower end of the size spectrum of nanometer size in diameter" somehow seemed a strange wording. The term "lower end" seeemed odd – suggest to replace that text above with something more linked to their formation process, replacing "at the lower..." with "freshly nucleated particles are observed at nanometer sizes..." then the rest of the sentence can continue with "as they grow....". Then similarly instead of "upper end of the spectrum" suggest "coarse part of the spectrum".

12) Introduction, page 2, line 9 – add ", which can also" before "affect rain formation" as it's not just the coarse mode particles which affect rain formation, the sub-micron ones do too – indeed the nanometre ones can too as long as they have enough time/added-condensating-vapours to grow them up to cloud-droplet-nucleating sizes.

13) Introduction, page 2, lines 11-12 – change the start of this setence to be more specific about the size effect you are explaining – in simple terms it can be understood simply as particles only interacting effectively with the radiation once they're above a certain size. I'd suggest to re-word the sentence to something like "There is a steep size dependence for how effectively aerosol particles interact with radiation (Chung et al., 2005) and clods (Lohmann and Feichter, 2005)." Suggest also to cite the chapters 7 and 8 of the 2013 IPCC AR5 report rather than the 2005 references given there – i.e. Myhre et al. (2013) and Boucher et al. (2013).

14) Introduction, page 2, line 13 – suggest to add qualifier after "entire aerosol size spectrum" as "(from nm to 10s of microns)" and please add "particle" between "aerosol" and "size spectrum".

15) Introduction, page 2, line 14 – delete "i.e." from that sentence, it makes more sense without that abbreviation.

16) Introduction, page 2, line 15 – replace "of the aerosol constituents, also influence" with "vary strongly between different aerosol constituents, including", and replace "cloud processes in the atmosphere" with "cloud interactions" (it's clearer then).

17) Introduction, page 2, line 18 – replace "describe the atmospheric aerosol" with "describe aerosol particles".

18) Introduction, page 2, line 18 – insert "particle" between "aerosol" and "size distribution".

19) Introduction, page 2, lines 19-20 – suggest to delete the word "detailed" from this sentence and replace "is not computationally feasible" with "is computationally challenging". The thing is that the word "detailed" might be understood differently by different people so an absolute yes/no to feasibility is not appropriate.

20) Introduction, page 2, line 23 – add "e.g." before "(Mann et al., 2014)".

21) Introduction, page 2, line 25 – replace "in size classes" with "into size classes".

22) Introduction, page 2, line 28 – there is also the Piecewise Lognormal Approximation (von Salzen, 2006) which has each size section represented as a log-normal distribution. Please add that as another approach here.

23) Introduction, page 3, line 1 – need to be more careful with this explanation here. Suggest to re-word the end of this sentence instead to say "the application of sectional models in global 3-D simulations often involves a trade-off with horizontal or vertical resolution" or similar.

24) Introduction, page 3, line 2 – replace "This is mainly because..." with "It is also hard to quantify the benefit of the sectional approach because...".

25) Introduction, page 3, line 6 – replace "a given parameter may not represent the observed value at a particular measurement site" with "a given observable may not represent the measured value at a particular monitoring site". The word "parameter" is not quite right there – I think the text above better reflects what you are trying to say there.

26) Introduction, page 3, line 12 – replace "This paper" with "The paper".

27) Introduction, page 3, line 14 – replace "simulations that were made" with "to be analysed" and replace "with different model configurations" with "with the different models/configurations". Also replace "present the evaluation" with "present an evaluation".

28) Introduction, page 3, line 15 – replace "radiative properties" with "optical properties".

29) Introduction, page 3, lines 16-17 – replace "in situ observations as well as aircraft observations of aerosol composition and mass." – the aircraft observations are in-situ measurements – suggest to re-word as: "in-situ observations, including vertical profiles of aerosol composition and mass from aircraft measurements."

30) Section 2.2. – lines 11-13 – The paper has not quite explained what is the distinction between HAM and M7. Until reading this I thought they were the same thing but I think I now understand that "HAM" is the overall aerosol module (including emissions, dry deposition, scavenging etc.) whereas M7 is just the aerosol microphysical routines. Am I understanding that correctly? If so this needs to be stated explicitly somewhere here – in so-doing it will help ensure the community apply the acronyms correctly and consistently in future.

31) Page 5, section 2.2 – line 20 – suggest to replace "represents several real-life compounds" with "represents several specific single-species compounds" if that is what is intended?

32) Page 5, section 2.2 – line 22 – suggest to replace "using compound classes" with "using lumped components" then replace "compounds" with "components" later in the sentence. Also on page 5 – replace "compounds" with "components".

33) Page 6, Table 1 – need to add additional entries to the "emissions" section for primary carbonaceous – and give the different size assumptions for emitted primary carbonaceous particles from biomass burning, bio-fuel and fossil-fuel sectors. As per my major comment 1, I think this is the primary reason why there is the AOD different in

those strong emissions regions. You can see that the BC size distribution is at different sizes in the sectional and modal scheme, and I think this can simply be explained by a different size assumption – I would be very suprised if that was caused by microphysical processing.

34) Page 7, line 7 – be clear what you mean by "coupled" – you mean "radiatively-coupled" right? Need to add an extra sentence briefly explaining how that's done here for aerosol-radiation interactions and aerosol-cloud interactions in the sectional scheme (and how that differs from the radiative coupling when the modal scheme is used).

35) Page 7, line 10 – you write "we used the climatologies" but I don't think you mean climatologies here do you? What is the time-variation of the specified SST and sea-ice distributions?

36) Page 8, line 12 – you write "using 3-dimensoinal fields from the MOZART..". Are these monthly-mean fields? If so please insert "monthly-mean" before "3-dimensional".

37) Page 8, line 16 – you write "For most of the processes the difference is only in the numerical treatment" but that's not quite right – the nucleation processes are different as shown in Table 1 – please change this wording.

38) Page 8, line 25 – you write "more detailed size-dependent scavenging rates" but you need to add a few qualifying words so the reader knows what you mean by "more detailed" here. The reader might expect the sectional SALSA scheme to have more detailed scavenging than the modal scheme – or maybe you don't mean detailed in a size-resolved way – do you mean the way the scavenging applies different scavenging efficiency for the different types of precipitating cloud?

39) Page 8, line 31 – suggest to replace "solving" with "kinetically, within the".

40) Page 9, line 6 – insert "2D (Herzog et al., 2004; Weisenstein et al., 2007) and box model (Kokkola et al., 2009)" between "Previous" and "studies" and then delete those

references from the end of the line.

41) Page 9, line 8 – replace "stratospheric solar radiation management by injecting sulfur into the stratosphere" with "stratospheric sulfur solar radiation management".

42) Page 9, line 9 – As per my major comment 2, it is not fair to refer to the initial settings of the scheme as "The default settings". They are indeed the default settings for tropospheric aerosol simulations, but they are not the default settings for stratospheric aerosol simulations. As per my major comment 2 please change the branding of these two M7 simulations from "M7" and M7mod" to "M7-trop" and "M7-strat". They are alternative configurations of M7 specifically for those applications. It's fine to show that simulation with the tropospheric configuration of M7 – in fact that will show why it's important to only apply the scheme in the stratosphere with the stratospheric configuration (M7-strat). But you need to change the wording so that it's clear that this is only default for tropospheric aerosol application of the model.

43) Page 9, line 11 – The authors write "This is because the high concentration of sulfur produces a bi-modal aerosol population". Is this statement referring to the Laramie balloon-borne OPC measurements (Deshler et al., 2003) which show the bimodal size distribution after Pinatubo? If so please give that reference here.

44) Page 9, line 12 – the narrowing of the width – again I think you are referring to what is observed from the measurements right? That is the case that the accumulation mode is observed to have a narrower size distribution – cite Deshler et al. (2003) or Deshler (2008).

45) Page 9, line 14-16 – you're referring to the box model simulations here, right? It's not so clear how the effect plays out in 3D simulations, and more so when you consider the trade-off in the better stratospheric circulation that can be afforded (by resolving more vertical levels for example) with a computationally faster aerosol scheme. So you need to be clear that you're referring here to here is what is seen in a box model. For a balanced discussion of this, you also need to add a qualifying sentence explaining

this trade-off between the cost of the aerosol scheme and the cost of the atmosphere model.

46) Page 9, line 15 – you need to re-word "grows too fast and the particles are sedimented too fast". The box model shows that in those simulations the growth proceeds faster, but you do need to qualify the 2nd part with "which would result in particles sedimenting faster" or something like this. Since it has not really been demonstrated in global models you need to tone down the way that is described.

47) Page 9, line 17 – It is not appropriate to refer to this as "A work-around solution". The "code-owners" of the M7 scheme are clear in their publications that when the scheme is applied for stratospheric aerosol applications, the modal settings need to be configured differently than for tropospheric aerosol applications. That's not correct to refer to that as a work-around. Effectively the scheme is only "licensed" to be applied in the stratosphere if it has this adjustment to the modal settings. As per my major comment 2 this section needs to be re-worded to make this clear – in my strong opinion, for the reasons above, you should refer to the tropospheric aerosol and stratospheric aerosol configurations of M7, and label them as "M7-trop" and "M7-strat". That is then consistent with the way the owners of the adjusted scheme have re-configured the model to be applicable for the stratosphere.

48) Page 9, line 22 – the Guo et al. (2004) has the SO2 emissions range as 14 to 23 Tg of SO2 – you need to give that range (and any widening of that to include values from other publications).

49) Page 9, lines 23-24 – change "produced stratospheric aerosol that that persisted in the stratosphere for over 3 years" with "perturbed the stratospheric aerosol layer for over 3 years". The distinction is important because it may be that particles transported to above 35km evaporated and their sulphur transferred to other "younger particles" in the stratospheric aerosol layer.

50) Page 9, line 29 – Replace "The setup for the volcanic emission was identical to the

one was used.." by "The emissions settings are identical to that used by...".

References ————-

Boucher et al. (2013) "Clouds and Aerosol", chapter 7 of "Climate Change 2013: The Physical Science Basis. Contributions to the Fifth Assessment Report of the Intergovernmental Panel on Climate Change", Cambridge University Press, 2013.

Deshler et al. (2003) "Thirty years of in situ stratospheric aerosol size distribution measurements from Laramie, Wyoming (41N), using balloon-borne instruments", J. Geophys. Res., vol. 108, no. D5, 4167, doi:10.1029/2002JD002514, 2003.

Deshler (2008), "A review of global stratospheric aerosol: Measurements, importance, life cycle, and local stratospheric aerosol", Atmos. Res., vol. 90, pp. 223–232, 2008.

Herzog et al. (2004) "A dynamic aerosol module for global chemical transport models: Model description", J. Geophys. Res., vol. 109, D18202, doi:10.1029/2003JD004405, 2004.

Lee et al. (2011) "Emulation of a complex global aerosol model to quantify sensitivity to uncertain parameters", Atmos. Chem. Phys., 11, 12253–12273, 2011.

Myhre et al. (2013) "Anthropogenic and Natural Radiative Forcing", chapter 8 of "Climate Change 2013: The Physical Science Basis. Contributions to the Fifth Assessment Report of the Intergovernmental Panel on Climate Change", Cambridge University Press, 2013.

Von Salzen (2006), "Piecewise log-normal approximation of size distributions for aerosol modelling", Atmos. Chem. Phys., 6, 1351-1372, 2006.

---

## Author Comment (AC2) · 15 Jun 2018

**Reply to Referee #1**

We thank the referee for the comprehensive review of our manuscript. We will improve the manuscript following the points made by the referee. Below are the replies to Referee #1. Original comments are in italic.

1. *Performance of SALSA2.0 is compared to the ECHAM-HAMMOZ default (modal) aerosol module M7. Consequently, many of the differences in this work are attributed to the difference in the numerical treatment of the aerosol size distribution in modal or sectional schemes. Because of this, the manuscript does not really give an indication of how well SALSA2.0 performs compared to other sectional modules.*

   Comparing SALSA2.0 to other sectional models would be extremely interesting. However, to have a meaningful comparison of two aerosol modules, they should be compared within the framework of the same global atmospheric model. If the host atmospheric models are different, it is extremely challenging to distinguish if the differences in the simulated aerosol fields are caused by the differences in the aerosol modules or the atmospheric models. In another study (Saponaro et al. (2018) in preparation), we compare the ability of three aerosol models to simulate aerosol-cloud interactions, two of which (HAM-M7 and HAM-SALSA) share the same atmospheric model ECHAM, while the third aerosol module is in a different global model. In that study, we show that aerosol properties are much more similar for the aerosol modules which are in the same atmospheric model than for the aerosol module in a different atmospheric model. This conclusion is also supported by the study by Mann et al. (2014) where 12 aerosol models of different complexity were compared. Thus, conducting a meaningful comparison between SALSA and another sectional model would require excessive work and therefore it would not be possible within this study. We will mention this in the revised manuscript. On the other hand, SALSA has been (e.g., Mann et al., 2014; Tsigaridis et al., 2014; Kipling et al., 2016) and will be included in the international AEROCOM project model experiments, where models are evaluated against each other and against observations.

2. *The paper describes the difference between SALSA1 and SALSA2.0 well, but does not discuss the reasons why certain changes are made to the SALSA module. Please explain what the main problems with SALSA1 are and how these changes contribute to the improvement of the aerosol module.*

   The major change between SALSA1 and SALSA2.0 was how the concentrations of different chemical species in different sized particles are treated. In SALSA1, particles were separated into three subregions depending on the particle size. In the subregion for the largest particles (larger than 700 nm), only the particle number and sea salt, mineral dust, and a lumped "water soluble species" mass concentrations were tracked. This choice was done to keep the number of tracer variables to the minimum. However, this had two unwanted implications. First, the total amount of sulfate, organic aerosol, and black

[Figure]

Figure 1: Simulated size distributions of a volcano plume (see Kokkola et al. (2009) for details) using SALSA2.0, SALSA1, and an explicit aerosol microphysics model MAIA.

carbon was lost when the particles grew to sizes larger that 700 nm. This was a practical problem in AEROCOM experiments where the information of individual species was required (e.g. Kipling et al., 2016). Second, although in the troposphere microphysics have very little influence on the size of particles in the third subregion, when simulating volcanic eruptions or stratospheric solar radiation management, condensation can grow the largest particles. This problem caused the model to have problems in simulating the growth of particles in a volcano plume since the third region particles did not grow which resulted in underestimating the effective radius of the volcano plume (Kokkola et al., 2009). In SALSA2.0 we extended the second subregion to cover also the size classes of subregion three. This way we can track the concentrations of all chemical compounds and the growth by microphysical processing is not limited in the largest size range. Figure 1 demonstrates how SALSA2.0 can simulate the growth of large particles while SALSA1 underestimates their growth. We will add discussion on these aspects in the revised manuscript.

Another major change was changing the size distribution structure from the moving center method (Jacobson, 2005) to the hybrid bin method (Young, 1974; Chen and Lamb, 1994). As explained in the manuscript, the moving center method causes numerical artefacts (Mohs and Bowman, 2011). We also have seen in our simulations that moving center consistenly produces lower number concentrations. For example, in Bergman et al. (2012) study, SALSA1 significantly underestimated the number size distributions measured at EUSAARI sites whereas in this study we did not experience such underestimation. This underestimation of particle numbers in SALSA1 was analyzed to be partly caused by

[Figure]

Figure 2: Relative difference [%] between in AOD the hybrid bin and moving center method (Figure adapted from the presentation "Representing the Evolution of Aerosol Size Distribution in Atmospheric Models" by Harri Kokkola in the 2015 International Aerosol Modeling Algorithms Conference, UC Davis, CA).

the moving center method. In Figure 2 we show the relative difference AOD over the US between a model setups using the hybrid bin method and the moving center method. From the figure we can see that AOD is always lower when using the moving center method. We will add a brief discussion of this in the revised manuscript.

3. *Previous work (also referred to in this article) has shown that modal aerosol representations do not perform well in simulating stratospheric aerosol caused by volcanic eruptions and that sectional approaches yield far better results. It is therefor not surprising that SALSA2.0 performs better than M7, but how does it compare to e.g. SALSA1? Overall, the discussion of the Pinatubo simulation is thin and mainly addresses issues in M7. As it is now, it might be better to remove this section from the paper as the remainder is already a very comprehensive comparison to observations.*

The intention of this comparison is not only to evaluate if SALSA2.0 performs better than M7 but also to evaluate how well SALSA2.0 can simulate the evolution of the stratospheric aerosol that originates from the Mt Pinatubo plume. This comparison is also justified because the volcano plume is simulated with a sectional model and a modal model coupled to the same atmospheric model. The main result from this evaluation is that SALSA2.0 can simultaneously simulate both tropospheric and stratospheric aerosol without changing the setup. It has to also be noted that M7 and SALSA can be consider to be used for different types of studies. As a sectional model, SALSA would be a preferable choice for simulating stratospheric aerosol, volcano eruptions, and geoengineering. However, we will add simulation results from SALSA1 and will also add discussion on the differences between SALSA1 and SALSA2.0 simulated volcanic aerosol.

4. *The authors state that a size-resolved wet deposition scheme for SALSA2.0 is still under development. To make a fair comparison to M7 an older removal scheme is also used for those simulations. In my opinion, the quality of this paper would greatly improve if the new wet deposition scheme is included in this work. In the comparison to aircraft measurements in the results section it is also explicitly mentioned how the new wet deposition scheme would improve the results of the simulations with SALSA2.0.*

   In this paper, the focus in comparing M7 and SALSA is in comparing the modal and sectional approach and thus we kept the two model configurations as close as possible. The new wet deposition scheme of M7 is used in the upcoming paper by Tegen et al. (2018) which will provide a detailed evaluation of ECHAM-HAMMOZ with M7. The inclusion of the new wet deposition scheme would in SALSA also be a valuable addition, however, here we present the release version ECHAM6.3.0-HAM2.3-MOZ1.0 which does not include the new wet deposition scheme. The work on the new wet deposition scheme is underway, but it is by no means a trivial task to include such a new scheme in the model. Simulating wet deposition properly in a global scale model is a major challenge for the whole aerosol modelling community. The improved wet deposition schemes in M7 are for a modal scheme and can not be applied to SALSA2.0. In order to implement an improved sectional wet deposition scheme in SALSA2.0 will require extensive model development, testing, and analysis and could be a topic of an independent article (see e.g. Croft et al. (2009, 2010)). Consequently, it is beyond what can be done within this manuscript.

5. *Title / Page 8, line 9-13 Judging from the text, the MOZ module is not used in this work and HAMMOZ is reduced to HAM only. This is a bit misleading and causes confusion in the text. It is not clear what value the combination of HAM and MOZ has in this work. It should be clarified better what MOZ does in the model simulations for this work, otherwise it might be better to remove MOZ from the title.*

   We will clarify in the revised manuscript that MOZ is not used in these simulations. However, the model licence requires that when publishing results with the model, the full name ECHAM-HAMMOZ is used and the exact release reference is stated.

6. *Page 2, line 31 100 size classes is a bit of an exaggeration, in global models the number of size classes is usually (much) lower.*

   Here we refer to the study by Mann et al. (2014) where the GEOS-Chem-APM model is reportedly using 100 size classes.

7. *Page 5, line 21-23 Two subranges of SALSA1 are combined into one in SALSA2.0, what is the reason for this simplification and what are expected changes in the simulated aerosol size distribution?*

See our reply to the second referee comment.

8. *Page 5, line 25-26 The moving center method is replaced by the hybrid bin method. What are the downsides to this method as is was not used in SALSA1 before?*

The moving center method was not chosen for SALSA1 because of downsides of the hybrid bin method but because it seemed to be suitable for global scale models. Like all size distribution methods, hybrid bin method suffers from numerical diffusion (e.g. Dinh and Durran, 2012). However, in our model framework it improved the correspondence between the simulated and the observed aerosol size distributions.

9. *Page 7, line 8-9 In view of the importance of meteorology for e.g. dust emissions, what is the nudging time interval used in the simulations?*

The relaxation times for the nudging of the surface pressure, vorticity, and divergence are 24 h, 6 h, and 48 h, respectively. We will add this information in the revised manuscript.

10. *Page 8, line 14-34 The different parameterisations of SALSA2.0 and M7 are explained extensively in Section 2.3, but several of these are changed for the simulations. This is very confusing and makes large parts of previous section irrelevant. It would be better to describe what is actually used. Also, this means that this work presents results from a suboptimal model run and the full potential of the SALSA2.0 module in the ECHAM-HAMMOZ model is not shown.*

With respect to SALSA, the only parameterization that was changed was secondary organic aerosol (SOA) formation using volatility basis set (VBS). We will remove the row describing SOA for both SALSA and M7. This setup is the default for the model release. It may be suboptimal for simulations of SOA formation. However, using VBS increases computational time and significanlty increases model output to the extent that it may not be practical to use it for simulations that do not focus on organic aerosol.

11. *Page 9, line 26 How is this ensemble constructed? What is the difference between the 5 members?*

We have perturbed offline anthropogenic aerosol emissions by values of the order of $10^{-6}$ which is an insignificant number for emission strengths but due to the chaotic nature of the atmospheric model, changes the model dynamics sufficiently. This will be clarified in the revised manuscript.

12. *Page 13, line 6 Judging from this work, the implementation of the new Long et al. (2012) sea salt emission decreases the model performance, why was it introduced?*

This is only true regarding aerosol mass. An upcoming study by Tegen et al. (2018) will show that the comparison between the measured and modelled number size distribution improves going from Guelle et al. (2001) to Long et al. (2011) parameterization. In addition, they show that aerosol optical depth compares better with MODIS retrieved AOD's when using the Long et al. (2011) parameterization.

13. *Page 15, line 4 Here, the low AOD bias is (almost) completely attributed to the low SS emissions. Although this assumption is acceptable for the SH, there is also a strong bias in the NH high latitudes. Here, the low bias over the land masses cannot be attributed to sea salt only.*

    This is a very good point. The low bias at high latitudes can also be attributed to issues in aerosol transport. Bourgeois and Bey (2011) have shown that ECHAM-HAMMOZ underestimates the aerosol load over the high latitudes. We will add discussion on this in the revised manuscript.

14. *Page 15, line 13-14 How did you arrive to this conclusion?*

    We came to this conclusion comparing modelled and observed time series. However, we will remove this line from the revised manuscript since these results are not shown in the manuscript.

15. *Page 15, line 25 Why is this not mentioned in Section 4.1.1?*

    We will add discussion on MODIS biases to Section 4.1.1

16. *Section 4.2 Restructure section. Multiple species of multiple model runs are compared to multiple measurement networks, This already makes the discussion hard to read. I suggest a fixed format/structure in discussing the different species to help the reader.*

    We will follow the suggestion of the referee and change the structure to separate different species.

17. *Page 18, line 1-22 Include comparison results of M7 in discussion of SU/BC.*

    We will add M7 results for sulfate and black carbon in the revised manuscript.

18. *Page 19, line 9 Aerosol load over oceans is not low over SH subtropics.*

    We will rephrase this to state that AOD is low south of latitude 40°S.

19. *Page 19, line 15 If periods in observations and simulations are different, how are they collocated?*

    The monthly model values are constructed by averaging daily means only for days where an observation is available. Moreover each model monthly mean is spatially colocated to the location of the observation station (by bi-linear interpolation). We will clarify this in the revised manuscript.

20. *Page 22, line 20 Why are monthly mean model values used here? Having 3 hourly output, collocation can be greatly improved. Also, comparison for SU and OA is based on daily mean output. What is the reason for this inconsistent approach?*

    This was done to enable an easier comparison between earlier model experiments. For black carbon, we reproduced the AEROCOM experiment by Koch et al. (2009) which

used monthly mean values. For sulfate and organic aerosol, we reproduced the model evaluation by Heald et al. (2011) where daily values were used.

21. ***Page 25, line 5-6*** *This conclusion is too strong and drawn too quickly. It would be the case for the ARCTAS Spring and ARCPAC campaigns, but for the ARCTAS Summer, the wet deposition scheme barely influences the results for the lower part of the atmosphere where observations are available. Also, for the source regions, an increase due to the wet deposition scheme, would increase the already high bias in SALSA.*

This is a good point from the referee. This is purely speculation and we will remove this sentence from the revised manuscript.

22. ***Page 26, line 1*** *Difference was fairly small". Can this statement be quantified?*

We have quantified this for the revised manuscript as follows. "For most campaigns and height levels, the relative difference in black carbon mass concentration is less than $50\%$ and the shape of the vertical profiles are very similar, mostly overlapping each other. The largest difference is for black carbon in the CR-AVE campaign where the relative difference in the mass concentration is ˜$83\%$ in the lowest layer.

23. ***Section 6*** *In this section, it is explained why the section approach of M7 does not perform well in simulation the stratospheric aerosol burden resulting from a volcanic eruption. There is even a reference to a solution for this problem. Yet you dont incorporate this in your model and compare the performance of SALSA2.0 mainly to the simulation with the unadjusted M7 scheme. As a result, it is difficult to really judge how well SALSA2.0 performs in these simulations. It would be more interesting to see the performance of SALSA2.0 to other sectional aerosol modules or modal schemes that were properly adjusted.*

Implementing a modal structure with varying mode width would require rewriting major part of HAM and M7. In addition, it would increase the amount of tracer variables which would be undesired for M7 in which the number of these variables is kept to the minimum. As it is said in the manuscript, the mode widths of M7 have been optimized for the troposphere and were not designed to represent stratospheric aerosol.

Regarding comparison to another sectional model, a meaningful comparison to another sectional model would have been too challenging to do within this study (as was explained above).

24. ***Page 26, line 1-2*** *How are the model values and observations collocated?*

In the simulations, where we used only year 2010 values, for black carbon, we used the modelled monthly values from the flight path for the corresponding month of the year. For sulfate and organic aerosol, we used the modelled daily values from the flight path for the day of the year which corresponded to the observations. For the whole period, we

used the the same values, but also for the corresponding year of the observations. We will clarify this in the revised manuscript.

25. **Section 7. Conclusions** *The structure of the conclusion section is unnecessarily confusing. Follow same order as discussion in Results section.*

    We will restructure Conclusions as suggested

26. **Page 29, line 6** *What are recommendations for optimizing?*

    The details will be explained in an upcoming study by Tegen et al. (2018)

27. **Page 30, line 1-2** *Underestimation of particle number in SALSA1 not mentioned in Section 4.1.1.*

    Here we refer to a study by Bergman et al. (2012) and will add the reference in the revised version

28. **3 Technical comments Page 4, line 12** *Sentence is not clear, please rephrase.*

    We will rephrase this as: "In its default setup, HAM uses the modal approach together with the model aerosol microphysics module M7 (Vignati et al., 2004).

29. **Page 5, line 13** *"using the volume ratio" → "using volume ratio"*

    This will be corrected in the revised manuscript

30. **Page 7, line 10** *Add full name of PCMDI*

    We will add the full name in the revised manuscript.

31. **Page 10, line 21-23** *Why is India omitted from this list?*

    We will add India to the list.

32. **Page 11, Figure 2** *Add statistics (e.g. corresponding global mean AOD to a,b,c and correlation coefficients and NMBs to d,e) to the figure for a good overview between model configurations.*

    We will do as suggested.

33. **Page 12, line 2** *Equation straightforward, can be removed and explained in words.*

    We will remove the equation.

34. **Page 13, line 1** *Fig 4. → Fig. 5*

    We will correct this.

35. **Page 14, line 12** *Reference to current section.*

    We will remove the reference.

36. ***Page 15, line 17*** *Add minus sign to 0.05.*

   We will correct this.

37. ***Page 16, Figure 6*** *Change colours of Asia and North America. These are the two regions discussed in the text but hardly distinguishable from each other. Also, adding a regional mean values would provide a good overview of model performance.*

   We will add regional values to the revised manuscript. However, to us the colors look sufficiently different and since the values for the two regions are completely different, they are easy to separate. We do not see the need for changing the colors.

38. ***Page 17, Figure 7*** *Remove additional abbreviations of species in lower left corner of each panel. Add names of network to panels.*

   We will do as suggested.

39. ***Page 20, Figure 9*** *Observed and simulated values in year 2010? Please add to caption.*

   This is correct and will be added to the figure caption.

40. ***Page 22, line 32*** *Vertical profiles of AMMA, ARCTAS Spring and OP3 are not captured well either.*

   We will mention this in the revised manuscript.

41. ***Page 26, line 1-2*** *Add references for the HIRS and lidar observations.*

   We will add the references.

42. ***Page 29, Fig. 14*** *Add errorbars to observed values.*

   We will add the uncertainty estimates of the observations in the figure caption.

43. ***Page 30, line 23*** *confiburations → configurations*

   We will correct this.

**References**

[revised manuscript text omitted]

---

## Author Comment (AC3) · 15 Jun 2018

**Reply to Referee #2**

We thank the referee for a comprehensive review of our manuscript. Addressing the points made by the referee will improve the paper. Below are the replies to Referee #2. Original comments are in italic.

1. *However, some of the principal statements made in the Abstract are not supported by the results presented in the paper, and much more care needs to be taken in the statements interpreting potential reasons for differences between the sectional and modal schemes.*

   *In particular, the authors claim that the size distribution comparisons shown in Figure 4, shown for locations where AOD difference is largest, are indicative that of different microphysical processing in the modal and sectional schemes, and that this is the reason why the two aerosol schemes predict different extinction/AOD in these regions. But from careful inspection of Figure 4, its clear that is not the case. The authors already identify the two locations (in China and in Russia) as regions where the observed AOD is very high, and clarify that the Russian site is in a region where biomass burning emissions are high. The China site is in a region of strong anthropogenic emissions. The size distribution of the black carbon (the black bars in the stacked bar chart) are very different between the SALSA and M7 simulations at both locations and this clearly indicates that there is a systematic difference in the size at which primary carbonaceous aerosol particles are emitted, which is a much more likely explanation of the reason for the difference.*

   *At the China site, the M7 run has about half of the BC in particles larger than 200nm, whereas for the SALSA run this is only about 10site, indicating that there is a systematic difference between the two schemes in the sizes assumed for primary carbonaceous emissions. This is an important issue, because if such large differences in AOD could indeed be attributed to the simpler modal scheme having inadequate representation of microphysical processing compared to the bin scheme, then this could be cited in the literature extensively as a reason to justify the additional expense required for sectional aerosol schemes.*

   *It is noticeable that whereas the Table 1 explains in detail the size segregation of the sea-salt and dust emissions, there is no information given about the assumed size at which the carbonaceous particles are emitted (yet these are the dominant primary aerosol in polluted regions).*

   *I am sure this is just an oversight in the writing of the paper, and that there was no intention to omit this information, or to make a statement that is not supported by the results.*

   *1) Abstract, page 1, lines 5-6 As per my main comments above about Figure 4, this sentence is not supported by the results and needs to be removed or reworded. If the authors can repeat either the M7 or the SALSA simulations with the emissions size distribution for emitted carbonaceous particles identical in the two schemes then it may be possible to make some statement about this, but the different BC-size-distribution in M7 clearly*

*indicates there is a substantial differece in the "emissions size distribution" applied for carbonaceous particles in the two aerosol schemes, which is much more likely reason for the difference in AOD between the two schemes. In any case the locations shown in Figure 4 (in Russia and China) are regions of very strong primary emissions. Lee et al. (2011) apply a perturbed parameter uncertainty analysis to show how (at least for the global aerosol microphysics scheme applied there) the regions where aerosol properties are most dominated by uncertainties in microphysical pro- cesses are away from such "emissions hot spots". So even if one of the models was re-run with the same "primary emissions size distribution" as for the other, one might expect any difference from microphysical processes to have most impact in a different region than the two locations shown.*

It is true and a good point that differences in emission sizes could explain the differences over areas with high anthropogenic emissions. However, in our simulations this is not the case since for offline anthropogenic emissions, we use identical emission size distributions for SALSA and M7 (see Page 7 in the manuscript). There will be some difference resulting from remapping modal emissions to SALSA size classes, however the emission masses and numbers and their size distributions are identical for M7 and SALSA. The significance of microphysics calculation over the chosen areas can be demostrated by two SALSA runs: one where condensing organics are treated either assuming them to be non-volatile and one where they are assumed to be semi-volatile, however so that the resulting secondary organic aerosol yield is approximately the same. In Figure 1 we show how the simulated yearly mean aerosol optical depth (AOD) compare against MODIS (Figure 1a) over China. Panel b shows the AOD with non-volatile OA and c with semi-volatile OA. From the figure, we can see that although everything else except for the microphysical treatment of OA is exactly the same, there is a large difference in simulate AOD's. Other size dependent processes such as wet removal do also contribute to the difference, but the changes are initiated by differences in microphysics.

[Figure]

a            b            c

Figure 1: Retrieved aerosol optical depth by a) MODIS, b) SALSA2.0 with non-volatile organic aerosol, and c) SALSA2.0 with semi-volatile organic aerosol.

Lee et al. (2011) studied the sensitivity of CCN on different microphysical processes. The conclusions from that study do not necessarily apply for AOD. It may also be that the sensitivities of a modal model such as GLOMAP-mode (which was used by Lee et al. (2011)) and a sectional model can be different. We have shown previously that mode merging causes some damping effect on the sensitivity of CCN sized particles (Korhola et al., 2014)

We will add a more detailed description of the emission sizes in the revised manuscript. However, we don't see a reason to change our conclusions on this matter.

2. *It is also clear from Figure 2 that, for many regions, the M7 scheme could actually be argued to perform better (compared to the MODIS AOD) than the SALSA scheme. SALSA seems to have substantial bias over North Africa and over marine regions in the Southern tropics, for example.*

This is true and we will add regional values for the comparison between the modeled and AERONET AOD for both SALSA and M7 for a more detailed comparison and discussion.

3. *One final general comment was that it needs to be stated somewhere early on in the text the difference between the acronyms "HAM" and "M7". My understanding is that "HAM" is the overall modal aerosol scheme and that M7 is a component of HAM, basically the modal microphysical routines.*

*The reason I ask for this clarification is that I was expecting then the SALSA to not just be an alternative to M7, but an alternative to HAM, and that perhaps the correct naming should then be ECHAM6.3.0-SALSA2.0-MOZ1.0 when the SALSA scheme is applied.*

*Please can the authors clarify I am understanding this correctly.*

*However, perhaps that is not quite right and the implementation of SALSA into the model has in fact only implemented the microphysics routines within SALSA (or indeed that SALSA has always only been the microphysics routines). I realise that within HAM there is a separate acronym for the microphysics routines (M7) than the overall modal framework, which is known as HAM. By contrast many other aerosol schemes do not have this distinction and there is only one acronym for the overall aerosol module including both the microphysics routines and the other aspects (primary emissions, dry deposition, scavenging). The naming convention of the different parts of the model are important in this case as it helps the reader to appreciate which aspects of the HAM scheme have been retained in the implementation of SALSA.*

*I realise different groups will have different ways of naming their modules and Im not necessarily suggesting the SALSA group come up with a new acronym for the microphysics elements of SALSA. However I do think it needs to be stated somewhere in the section 2.2 description exactly what constitutes the Hamburg Aerosol Model and what are the SALSA aspects. See also my first specific comment about the wording of the title.*

You have understood the difference between HAM, M7 and SALSA correct. As explained in Section 2.2, HAM is the aerosol model which calculates aerosol emissions, removal, hydration, and radiative properties. M7 and SALSA are the aerosol microphysics modules of HAM which uses either modal or sectional approach depending on which aerosol microphysical module is used. Most of the actual HAM code is shared with both M7 and SALSA. This is why the model is called ECHAM-HAMMOZ. In addition, the model licence requires that when publishing results with the model, the full name ECHAM-HAMMOZ is used and the exact release reference is stated. We will further clarify this in Section 2.2.

*2) Abstract, page 1, lines 9-11 This difference in the modal and sectional aerosol microphysics predictions for the microphysical evolution and global dispersion of the Pinatubo volcanic cloud is interesting, but, as the authors point out, the standard mode widths for M7 are not intended to be applied to the stratospheric aerosol evolution. Figure 14 shows that actually, provided the model is applied with the "stratospheric- enabling adjustment" to the accumulation mode and coarse mode widths, the modal scheme compares well to the sectional scheme. As I understand it, the Hamburg stratospheric aerosol modelling group would not apply the model without this adjustment to the mode widths, so the emphasis really needs to be changed in how this is worded in the Abstract and in the discussion of the results. I think it is very important, to minimise the chance of an incorrect inference from the reader, to present the esults having that "M7mod" essentially as the default (or even "validated"?) configuration for when the model is applied for simulating interactive stratospheric aerosol. Indeed I would strongly recommend to change the "branding" of that model run to "M7-strat" rather than "M7-mod". As I understand it, the adjustment to the mode widths is a pre-requisite for simulating stratospheric aerosol for that scheme, so the authors of the manuscript need to change the current wording of the results to be clearer that it is essentially "the stratospheric configuration of M7" or so. One could consider it in some ways equivalent to a tropospheric or stratospheric chemistry scheme. One would not apply a tropospheric chemistry scheme to simulate the chemistry of the stratosphere.*

This is a very good point and we will add an explanation that the modified mode width used in the Pinatubo comparison is the stratospheric setup of M7, which is untested for the troposphere. We will also add references where it has been used: Niemeier et al. (2009); U. et al.; Niemeier and Timmreck (2015); Niemeier and Schmidt (2017). We will also change the model run name from M7mod to M7-strat following the suggestion of the referee. On the other hand, we need to emphasize that this setup is not a feature in this model release and using it requires code level changes.

**List of minor revisions**

Referee comments 3, 5, 6, 8, 9, 10, 12, 14-21, 24-29, 32, 36, 39-41, 49, and 50 suggest rewording and changes in the terms used. We will make the changes as the referee suggests in the revised manuscript.

*1) Title, page 1, The way the title is currently worded suggests the SALSA2.0 module is as a new sub-model within the Hamburg Aerosol Model (HAM). Is that the case then that HAM includes both M7 and SALSA as alternative aerosol microphysics modules? Or is the SALSA module an alternative to "the overall HAM" or so?*

SALSA2.0 is a new submodel and an alternative to M7 within HAM and a component of the release version of ECHAM-HAMMOZ. We will clarify this in the revision.

*2) Abstract, page 1, 1st sentence  Related to point 1) is ECHAM-HAMMOZ still ECHAM-HAMMOZ when SALSA is applied  or should it then be referred to as ECHAM-SALSAMOZ or so?*

As explained above, the model name remains ECHAM-HAMMOZ even with SALSA turned on.

*4) Abstract, page 1, 3rd sentence  insert "within ECHAM" or "within ECHAM- HAMMOZ" between "implementation" and "is evaluated" to be clear it is this particular implementation that is evaluated (one could imagine it potentially being implemented in another framework at some point in the future).*

We will insert "within ECHAM-HAMMOZ" in this sentence.

*7) Abstract, page 2, lines 1-2  as per major comment 1-2, this sentence needs to be changed since (as I understand it) the M7 microphysics would not be applied for stratospheric aerosol applications unless the mode widths for the accumulation and coarse soluble modes were reduced to 1.2 in this way. In this sentence and the results of the Pinatubo comparisons, I this should be referred to as "the stratospheric aerosol configuration of M7" or similar.*

We will refer to the M7 setup with modified mode widths as "the stratospheric aerosol configuration of M7" in the revised manuscript.

*11) Introduction, page 2, lines 8-9  The words "at the lower end of the size spectrum of nanometer size in diameter" somehow seemed a strange wording. The term "lower end" seeemed odd  suggest to replace that text above with something more linked to their formation process, replacing "at the lower..." with "freshly nucleated particles are observed at nanometer sizes..." then the rest of the sentence can continue with "as they grow....". Then similarly instead of "upper end of the spectrum" suggest "coarse part of the spectrum".*

It is true that the original wording is too complicated.  We will rephrase this to "For example, when the nanometer sized smallest partices grow in size, they contribute to the number of aerosol particles which can form cloud droplets (Kulmala and Kerminen, 2008) while the largest particles of micrometer size affect rain formation (Jensen and Lee, 2008)."

*13) Introduction, page 2, lines 11-12  change the start of this setence to be more specific about the size effect you are explaining  in simple terms it can be understood simply as*

*particles only interacting effectively with the radiation once theyre above a certain size. Id suggest to re-word the sentence to something like "There is a steep size dependence for how effectively aerosol particles interact with radiation (Chung et al., 2005) and clods (Lohmann and Feichter, 2005)." Suggest also to cite the chapters 7 and 8 of the 2013 IPCC AR5 report rather than the 2005 references given there i.e. Myhre et al. (2013) and Boucher et al. (2013).*

The size dependence of particles on radiation and cloud formation is so complex that we do not want to go into more details as it would require a lot of text to explain it comprehensively. This sentence was meant to only briefly mention the size dependency of these effects. We will add the suggested references.

*22) Introduction, page 2, line 28 there is also the Piecewise Lognormal Approxi- mation (von Salzen, 2006) which has each size section represented as a log-normal distribution. Please add that as another approach here.*

We will add mention this approach together with the reference in the revised version of the manuscript.

*23) Introduction, page 3, line 1 need to be more careful with this explanation here. Suggest to re-word the end of this sentence instead to say "the application of sectional models in global 3-D simulations often involves a trade-off with horizontal or vertical resolution" or similar.*

Here we mean to say that in global 3D models so many other processes than aerosol microphysics affect the atmospheric aerosol properties that the improvement due to a higher aerosol size resolution is not evident. We will clarify this in the revised manuscript. With SALSA we do not have a trade-off with horizontal or vertical resolution compared since they are the same for SALSA and M7.

*30) Section 2.2. lines 11-13 The paper has not quite explained what is the distinction between HAM and M7. Until reading this I thought they were the same thing but I think I now understand that "HAM" is the overall aerosol module (including emissions, dry deposition, scavenging etc.) whereas M7 is just the aerosol microphysical routines. Am I understanding that correctly? If so this needs to be stated explicitly somewhere here in so-doing it will help ensure the community apply the acronyms correctly and consistently in future.*

You have understood it correctly. It seems to be a common misunderstanding that M7 and HAM are synonymous. We will clarify the differences in the revised manuscripts.

*31) Page 5, section 2.2 line 20 suggest to replace "represents several real-life com- pounds" with "represents several specific single-species compounds" if that is what is intended?*

We will replace "real-life compounds" with "individual chemical compounds".

*33) Page 6, Table 1 need to add additional entries to the "emissions" section for primary carbonaceous and give the different size assumptions for emitted primary carbonaceous particles from biomass burning, bio-fuel and fossil-fuel sectors. As per my major comment 1, I think this is the primary reason why there is the AOD different in those strong emissions regions. You can see that the BC size distribution is at different sizes in the sectional and modal scheme, and I think this can simply be explained by a different size assumption I would be very suprised if that was caused by microphysical processing.*

We will describe more in detail the emission size distributions in the revised manuscript.

*34) Page 7, line 7 be clear what you mean by "coupled" you mean "radiatively-coupled" right? Need to add an extra sentence briefly explaining how thats done here for aerosol-radiation interactions and aerosol-cloud interactions in the sectional scheme (and how that differs from the radiative coupling when the modal scheme is used).*

Here we mean that we run ECHAM-HAMMOZ with SALSA2.0 aerosol microphysics. We will rephrase this accordingly in the revised manuscript.

*35) Page 7, line 10 you write "we used the climatologies" but I dont think you mean climatologies here do you? What is the time-variation of the specified SST and sea-ice distributions?*

We do use sea surface temperature and sea ice cover climatologies. They are monthly fields which we will clarify in the revised manuscript.

*37) Page 8, line 16 you write "For most of the processes the difference is only in the numerical treatment" but thats not quite right the nucleation processes are different as shown in Table 1 please change this wording.*

We will rephrase this part as follows: "In the default setups of M7 and SALSA2.0, wet deposition and secondary organic aerosol (SOA) formation are the only processes (in addition to the calculation of aerosol microphysics) that use different methods for solving the physics of the process. For the rest of the processes the difference is only in the numerical treatment.".

*38) Page 8, line 25 you write "more detailed size-dependent scavenging rates" but you need to add a few qualifying words so the reader knows what you mean by "more detailed" here. The reader might expect the sectional SALSA scheme to have more detailed scavenging than the modal scheme or maybe you dont mean detailed in a size-resolved way do you mean the way the scavenging applies different scavenging efficiency for the different types of precipitating cloud?*

We will modify this part to indicate that the wet deposition scheme is more physically based instead of using the ambiguous term "detailed".

*42) Page 9, line 9 As per my major comment 2, it is not fair to refer to the initial settings of the scheme as "The default settings". They are indeed the default settings for tropospheric aerosol simulations, but they are not the default settings for stratospheric*

*aerosol simulations. As per my major comment 2 please change the branding of these two M7 simulations from "M7" and M7mod" to "M7-trop" and "M7-strat". They are al- ternative configurations of M7 specifically for those applications. Its fine to show that simulation with the tropospheric configuration of M7 in fact that will show why its important to only apply the scheme in the stratosphere with the stratospheric configuration (M7-strat). But you need to change the wording so that its clear that this is only default for tropospheric aerosol application of the model.*

This is true and we will modify this part as explained in the reply to the major comments.

*43) Page 9, line 11 The authors write "This is because the high concentration of sulfur produces a bi-modal aerosol population". Is this statement referring to the Laramie balloon-borne OPC measurements (Deshler et al., 2003) which show the bimodal size distribution after Pinatubo? If so please give that reference here.*

Here we refer to the model study (Kokkola et al., 2009), but we will also include the Deshler reference.

*44) Page 9, line 12 the narrowing of the width again I think you are referring to what is observed from the measurements right? That is the case that the accumulation mode is observed to have a narrower size distribution cite Deshler et al. (2003) or Deshler (2008).*

Here we refer to the detailed aerosol microphysics model which was used in the Kokkola et al. (2009) paper.

*45) Page 9, line 14-16 youre referring to the box model simulations here, right? Its not so clear how the effect plays out in 3D simulations, and more so when you consider the trade-off in the better stratospheric circulation that can be afforded (by resolving more vertical levels for example) with a computationally faster aerosol scheme. So you need to be clear that youre referring here to here is what is seen in a box model. For a balanced discussion of this, you also need to add a qualifying sentence explaining this trade-off between the cost of the aerosol scheme and the cost of the atmosphere model.*

Yes, we are referring to the box-model study Kokkola et al. (2009). However, the same effect can be seen in our global simulations where we can see that the largest particles are removed faster in the tropospheric setup M7 and is evident in the manuscript Figure 14b.

The difference in the computational speed between M7 and SALSA is not so large that they would use different grid resolution. We will however mention the increase in the computational burden when using SALSA.

*46) Page 9, line 15 you need to re-word "grows too fast and the particles are sedimented too fast". The box model shows that in those simulations the growth proceeds faster, but you do need to qualify the 2nd part with "which would result in particles sedimenting faster" or something like this. Since it has not really been demonstrated in global models you need to tone down the way that is described.*

We will modify this sentence as suggested. However, we investigated the difference between the growth of particles between the two M7 simulations. With the default mode widths, the coarse particle burden grows quicker right after the eruption and these particles are consequently removed much faster than in the case of using narrower modes.

*47) Page 9, line 17  It is not appropriate to refer to this as "A work-around solution". The "code-owners" of the M7 scheme are clear in their publications that when the scheme is applied for stratospheric aerosol applications, the modal settings need to be configured differently than for tropospheric aerosol applications. Thats not correct to refer to that as a work-around. Effectively the scheme is only "licensed" to be applied in the stratosphere if it has this adjustment to the modal settings. As per my major comment 2 this section needs to be re-worded to make this clear  in my strong opinion, for the reasons above, you should refer to the tropospheric aerosol and stratospheric aerosol configurations of M7, and label them as "M7-trop" and "M7-strat". That is then consistent with the way the owners of the adjusted scheme have re-configured the model to be applicable for the stratosphere.*

We will change the label to M7-strat. However, it has to be noted that the model version we present here requires code level changes to start using the alternative stratospheric aerosol configuration. It also has to be noted that there is no release version of ECHAM-HAMMOZ, which would support easy switch to this configuration. We will also remove the term "work-around".

*48) Page 9, line 22  the Guo et al. (2004) has the SO2 emissions range as 14 to 23 Tg of SO2  you need to give that range (and any widening of that to include values from other publications).*

We will give this range in the revised manuscript.

**References**

Kokkola, H., Hommel, R., Kazil, J., Niemeier, U., Partanen, A.-I., Feichter, J., and Timmreck, C.: Aerosol microphysics modules in the framework of the ECHAM5 climate model - intercomparison under stratospheric conditions, Geosci. Model Dev., 2, 97–112, https://doi.org/10.5194/gmd-2-97-2009, URL http://www.geosci-model-dev.net/2/97/2009, 2009.

Korhola, T., Kokkola, H., Korhonen, H., Partanen, A.-I., Laaksonen, A., Lehtinen, K. E. J., and Romakkaniemi, S.: Reallocation in modal aerosol models: impacts on predicting aerosol radiative effects, Geosci. Model Dev., 7, 161–174, https://doi.org/10.5194/gmd-7-161-2014, URL https://www.geosci-model-dev.net/7/161/2014/, 2014.

Lee, L. A., Carslaw, K. S., Pringle, K. J., Mann, G. W., and Spracklen, D. V.: Emulation of a complex global aerosol model to quantify sensitivity to uncertain parameters, Atmospheric Chemistry and Physics, 11, 12 253–12 273, https://doi.org/10.5194/acp-11-12253-2011, URL `https://www.atmos-chem-phys.net/11/12253/2011/`, 2011.

Niemeier, U. and Schmidt, H.: Changing transport processes in the stratosphere by radiative heating of sulfate aerosols, Atmos. Chem. Phys, 17, 14 871–14 886, https://doi.org/10.5194/acp-17-14871-2017, URL `https://www.atmos-chem-phys.net/17/14871/2017/`, 2017.

Niemeier, U. and Timmreck, C.: What is the limit of climate engineering by stratospheric injection of $SO_2$?, Atmos. Chem. Phys, 15, 9129–9141, https://doi.org/10.5194/acp-15-9129-2015, URL `https://www.atmos-chem-phys.net/15/9129/2015/`, 2015.

Niemeier, U., Timmreck, C., Graf, H.-F., Kinne, S., Rast, S., and Self, S.: Initial fate of fine ash and sulfur from large volcanic eruptions, Atmos. Chem. Phys, 9, 9043–9057, https://doi.org/10.5194/acp-9-9043-2009, URL `https://www.atmos-chem-phys.net/9/9043/2009/`, 2009.

U., N., H., S., and C., T.: The dependency of geoengineered sulfate aerosol on the emission strategy, Atmospheric Science Letters, 12, 189–194, https://doi.org/10.1002/asl.304, URL `https://rmets.onlinelibrary.wiley.com/doi/abs/10.1002/asl.304`.

---

## Referee Report (RR1)

Review of "SALSA2.0: The sectional aerosol module of the aerosol-chemistry-climate model ECHAM6.3.0-HAM2.3-MOZ1.0 by Harri Kokkola, Thomas Kuehn, Anton Laakso et al.,

At the end of May 2018, I reviewed the GMD-Discussions version of this manuscript, which describes the implementation of a sectional aerosol microphysics module (SALSA2.0) within the composition-climate model ECHAM-HAMMOZ (ECHAM6.3.0-HAM2.3-MOZ1.0), as an alternative to the existing modal aerosol microphysics scheme "M7".

My review identified the paper certainly within scope of GMD and found valuable the evaluation of aerosol optical properties, aerosol mass and particle size distribution simulated by ECHAM-HAMMOZ-SALSA, comparing to observations and parallel similar simulations with the existing aerosol scheme (ECHAM-HAMMOZ-M7).

However, I contended that some of the principal statements made in the Abstract were not supported by the results presented in the paper, and that more care needed to be taken in the statements interpreting potential reasons for differences between the sectional and modal schemes.

In particular, the authors claimed, and continue to claim in this revised version, that the size distribution comparisons shown in Figure 4, shown for locations where AOD difference is largest, are caused by different microphysical processing in the modal and sectional schemes, and that this is the reason why the two aerosol schemes predict different extinction/AOD in these regions.

My review pointed out that, as currently worded, readers of the paper might, not unreasonably, infer from this statement (and the associated results) that applying modal aerosol microphysics schemes in global models might have some large systematic error compared to sectional schemes which could be considered a benchmark test for the model.

This potential general inference therefore requires the authors to make sure their statements are fully supported by the results.

In my review of the GMD-D paper, I pointed out that the two cases shown in Figure 4 (which are in locations where primary aerosol emissions are very high, of predominantly biomass burning emissions and of anthropogenic emissions), the size distribution of the black carbon (the black bars in the stacked bar chart) are very different between the SALSA and M7 simulations at both locations.

And that, to me, this clearly suggested that there was likely a systematic difference in the size at which primary carbonaceous aerosol particles are emitted, which could explain the reason for the difference. For example at the China site, the M7 run has about half of the BC in particles larger than 200nm, whereas for the SALSA run this is only about 10%. The same is true for the Russia site, indicating that there seems to be a systematic difference between the two schemes in the sizes assumed for primary emitted carbonaceous particles.

The authors replied to this as follows:

"It is true and a good point that differences in emission sizes could explain the differences over areas with high anthropogenic emissions. However, in our simulations this is not the case since for offline anthropogenic emissions, we use identical emission size distributions for SALSA and M7 (see Page 7 in the manuscript). There will be some difference resulting from remapping modal emissions to SALSA size classes, however the emission masses and numbers and their size distributions are identical for M7 and SALSA."

So the authors agree that if there were differences in the primary emitted size distributions (i.e. differences before the aerosol schemes transported the particles) then this could explain the differences seen.

But they explain that the assumptions for the emissions are identical, so unless there is a systematic difference introduced when mapping modal emissions to SALSA size classes, this is not the cause of the difference.

However, in their reply to my review, the authors go on to clarify that they are actually attributing the cause of the differences to differences in the way secondary organic aerosol is treated in the model – and that that is what they mean by "differences in microphysical processing".

Specifically they explain:

"The significance of microphysics calculation over the chosen areas can be demonstrated by two SALSA runs: one where condensing organics are treated either assuming them to be non-volatile and one where they are assumed to be semi-volatile, however so that the resulting secondary organic aerosol yield is approximately the same."

This information is extremely interesting in that it clearly demonstrates that the way models represent the size-resolved partitioning of organic aerosol has a major impact on simulated aerosol optical properties.

However, again the authors are making an incorrect statement in contending that this demonstrates a difference arising from the modal or sectional aerosol microphysics (the aerosol dynamics approach).

Certainly this shows the major significance/influence from the way gas to particle partitioning is applied in the model, but it does not give any information about any apparent difference between modal and sectional aerosol microphysics schemes.

In fact this statement then strengthens the points I made in my original review, that the differences between the ECHAM-HAM and ECHAM-SALSA are not caused by differences between the modal and sectional schemes, but by differences in the treatment of organic aerosol.

The additional Figure 1 added in the reply to reviewers is very interesting as it shows that treating the semi-volatile nature of the organic aerosol

has a major control on simulated aerosol optical properties.

I agree completely with the authors' statement in the text just above that Figure 1 when they explain:

From the figure, we can see that although everything else except for the microphysical treatment of OA is exactly the same, there is a large difference in simulate AOD's

And I also agree that this also highlights the importance of aerosol microphysical processes.

But in the revised manuscript, the Abstract still states:

The largest differences between SALSA2.0 and M7 are evident over regions where the aerosol size distribution is heavily modified by the microphysical processing of aerosol particles

I am still not content to approve this manuscript until the text clearly explains this is not a difference between the modal and sectional aerosol schemes, but a difference in the treatment of organic aerosol.

As currently worded, there is still a danger that a time-pressed reader, who may (not unreasonably) have to speed-read the manuscript for its main points, may incorrectly infer that the "difference in microphysical processing" between the ECHAM-SALSA and ECHAM-M7 simulations is due to a difference arising from the parameterized (modal) aerosol dynamics in M7 compared to the sectional aerosol dynamics applied in SALSA.

In their reply to reviewers the authors have shown that although it is correct to say that there is a difference in microphysical processing between the ECHAM-SALSA and ECHAM-M7 runs, their reply clarifies that the difference in simulated aerosol properties arise from the different treatments of organic aerosol between the two aerosol schemes, not from their different aerosol dynamics schemes.

To protect against this incorrect inference, it may simply be a case of revising the statements that attribute the cause of the differences to specifically identify the reason for the different simulated aerosol properties as the gas-particle partitioning of the organic aerosol (rather than the modal/sectional aerosol dynamics).

I request that the authors go through the manuscript and make sure the text cannot be misinterpreted by the time-pressed reader.

I can confirm that I am fully prepared to review the manuscript again once the text has been revised to clarify the attribution of the differences in simulated aerosol properties between the ECHAM-SALSA and ECHAM-M7 runs.

Finally, please note that whereas I am again classifying this required change as a major revision in my reviewers report, I am fully supportive of the manuscript being published in GMD once the required changes have been made. Furthermore I am confident the authors can make the changes

fairly quickly so am confident this additional revision will not introduce a significant delay in the review process.

---

## Author Response (AR2)

We have modified the manuscript addressing the comment from Referee #2 who required further changes to the manuscript. In addition, Editor had additional comments. Below are the replies to comments followed by the revised manuscript which includes all the changes marked in blue.

**Reply to Referee #2**

*"I am still not content to approve this manuscript until the text clearly explains this is not a difference between the modal and sectional aerosol schemes, but a difference in the treatment of organic aerosol.*

*As currently worded, there is still a danger that a time-pressed reader, who may (not unreasonably) have to speed-read the manuscript for its main points, may incorrectly infer that the difference in microphysical processing between the ECHAM-SALSA and ECHAM-M7 simulations is due to a difference arising from the parameterized (modal) aerosol dynamics in M7 compared to the sectional aerosol dynamics applied in SALSA.*

*In their reply to reviewers the authors have shown that although it is correct to say that there is a difference in microphysical processing between the ECHAM-SALSA and ECHAM-M7 runs, their reply clarifies that the difference in simulated aerosol properties arise from the different treatments of organic aerosol between the two aerosol schemes, not from their different aerosol dynamics schemes. To protect against this incorrect inference, it may simply be a case of*

*revising the statements that attribute the cause of the differences to specifically identify the reason for the different simulated aerosol properties as the gas-particle partitioning of the organic aerosol (rather than the modal/sectional aerosol dynamics).*

*I request that the authors go through the manuscript and make sure the text cannot be misinterpreted by the time-pressed reader."*

Thank you for this comment. We have tried to improve the text based on your suggestion and have revised the manuscript to emphasize that the differences between SALSA and M7 can very likely be explained by the differences in the methods used in the microphysics modules. The following changes were made in the text:

**Page 1, Abstract, Line 5, we now state that:** *The largest differences between the implementation of SALSA2.0 and M7 are in the methods used for calculating microphysical processes, i.e. nucleation, condensation, coagulation, and hydration. These differences in the microphysics are reflected in the results so that the largest differences between SALSA2.0 and M7 are evident over regions where the aerosol size distribution is heavily modified by the microphysical processing of aerosol particles.*

**Page 13, Line 3, we have modified the beginning of the paragraph to read:** *When*

*analyzing the aerosol mass size distributions, it is evident that over these locations the aerosol extinction is strongly affected by the differences between SALSA2.0 and M7 in the methods used for calculating microphysical processes, especially gas-to-particle partitioning. For calculating concurrent nucleation and condensation, M7 uses the method introduced by Kokkola et al. (2009) and SALSA2.0 the method by Jacobson (2005).*

**Page 16, Line 22, we have replaced** *microphysics* **by** *the treatment of microphysical processes, especially gas-to-particle partitioning,*

**Reply to Editor**
*Thank you for submitting a revised version. I sent this version to the two initial reviewers who kindly took a look. I agree with the reviwers' assessment that some of their initial comments have not been fully addressed in the revised version. For example, the comparison between SALSA2.0 and M7 as it is now does not seem very helpful for the reader to decide whether SALSA2.0 is really an improvement over SALSA1. I agree with the reviewer that it is worth adressing this question as it is important for anyone considering to use SALSA1/2.0. Please also see more comments attached (pdf). Please address all of these comments in the second revised version before publication in GMD can be considered.*

Thank you for this comment! Please note that we did clarify the shortcomings SALSA1 (and why changed some of the methods in SALSA2.0) in the previous round of the review as follows:

*The change in how the chemical compounds are treated was first of all due to practical reasons. In SALSA1, the information of individual species was lost when the particles grew to sizes larger 30 than 700 nm in diameter. This caused problems in studies where the information of individual species in all particle sizes was required (e.g. Kipling et al., 2016). Second, although in the troposphere microphysics have very little influence on the size of particles in the third subregion, when simulating volcanic eruptions or stratospheric solar radiation management, condensation can grow the largest particles. This caused the model to have problems in simulating the growth of particles in a volcano plume since the third region particles did not grow. This in turn resulted in underestimation of the effective radius of the volcano plume (Kokkola et al., 2009). In 0-dimensional model tests (not shown here), SALSA2.0 did not exhibit such problems.*

It has to be noted that SALSA1 is not an option for ECHAM-HAMMOZ and thus its use cannot be considered to be used without a substantial effort in reprogramming the coupling. To emphasize this, we have added the following line in the revised manuscript (Page 6, Line 1): *
[revised manuscript text omitted]